# Event Tables for Efficient Experience Replay

**Varun Kompella**  *varun.kompella@sony.com*
*Sony AI*

**Thomas J. Walsh**  *thomas.walsh@sony.com*
*Sony AI*

**Samuel Barrett**  *samuel.barrett@sony.com*
*Sony AI*

**Peter R. Wurman**  *peter.wurman@sony.com*
*Sony AI*

**Peter Stone**  *pstone@cs.utexas.edu*
*Sony AI*
*The University of Texas at Austin, Austin, TX 78712, USA*

**Reviewed on OpenReview:** *https://openreview.net/forum?id=XejzjAjKjv*

## Abstract

Experience replay (ER) is a crucial component of many deep reinforcement learning (RL) systems. However, uniform sampling from an ER buffer can lead to slow convergence and unstable asymptotic behaviors. This paper introduces Stratified Sampling from Event Tables (SSET), which partitions an ER buffer into Event Tables, each capturing important subsequences of optimal behavior. We prove a theoretical advantage over the traditional monolithic buffer approach and combine SSET with an existing prioritized sampling strategy to further improve learning speed and stability. Empirical results in challenging MiniGrid domains, benchmark RL environments, and a high-fidelity car racing simulator demonstrate the advantages and versatility of SSET over existing ER buffer sampling approaches.

## 1 Introduction

Many recent deep reinforcement learning (RL) breakthroughs (Mnih et al., 2013; Silver et al., 2016; Wurman et al., 2022) rely on Experience Replay (ER) and the corresponding buffer (an ERB) to store massive amounts of data that is re-sampled during training. Consider, however, a high-frequency car-racing simulator where an agent takes thousands of steps to complete a lap and where crucial *events*, such as passing another car, may occur on just a few of those steps. Uniform random sampling from an ERB populated with all the lap data is highly unlikely to focus on this key event. Prioritized Experience Replay (PER) (Schaul et al., 2016), which skews sampling based on TD Errors, might do better, but may also focus on states unlikely to be reached by the optimal policy (Oh et al., 2021). To address these limitations of existing ER methods, this paper introduces Event Tables, ERB partitions that hold sub-trajectories leading to events, and a corresponding wrapper algorithm, Stratified Sampling from Event Tables (SSET), to build training samples for off-policy RL.

In large domains, simply over-sampling the small number of disconnected state / actions where events occur is unlikely to be beneficial since initial state values would still rely on uniform sampling of the states between event occurrences. Instead, we take a lesson from previous works on trajectory-based backups (Barto et al., 1995; Karimpanal & Bouffanais, 2018) and store the finite-length history that preceded the event in the corresponding event table. Intuitively, this data forms a "fast lane" for backups between event occurrences

that chains back to the initial state(s). And by sampling individual steps from each table i.i.d., SSET avoids the instability (de Bruin et al., 2015; 2016) of using temporally correlated data in mini-batches.

We develop a theoretical underpinning for the fast-lane intuition and show that, if the events are correlated with optimal behavior and histories are sufficiently long, SSET can dramatically speed up the convergence of off-policy learning compared to using uniform sampling or even PER. Even if those conditions fail, a bias correction term preserves the Bellman target, although convergence may be slowed. From our empirical results, these properties translate to different off-policy RL base learners including DDQN (Van Hasselt et al., 2016), SAC (Haarnoja et al., 2018), and the distributional QR-SAC (Wurman et al., 2022) algorithm.

While SSET is a new way to optimize sampling from an ERB, it is complementary to many existing prioritization approaches or behavior shaping techniques. Specifically, SSET can be applied based on known events with TD-error PER used within each table, thereby focusing on crucial states that also need value updates. We apply this "best of both worlds" approach in many of our experiments and show that it performs better than using only one of the techniques. Similarly, SSET outperforms potential-based reward shaping in our empirical experiments, but the combination of two provides both better agent exploration and more efficient backups. Finally, viewing each event table as a data set for a particular skill, SSET can mitigate catastrophic forgetting (Goodfellow et al., 2014; Kirkpatrick et al., 2017) in RL. Our experiments show this advantage in acquiring multiple skills (Section 5.4) and retaining skills over time (Section 7.2).

This paper makes several contributions for ER using Event Tables. (1) We introduce Event Tables and the SSET framework. (2) We derive theoretical guarantees quantifying the sample complexity improvement with properly designed events and provide a bias correction that ensures the Bellman target is preserved. (3) We empirically demonstrate the advantages of SSET over uniform sampling or PER in challenging MiniGrid environments and continuous RL benchmarks (MuJoCo and Lunar Lander), and find that combining SSET with TD-error PER or potential-based reward shaping can further improve learning speed. (4) We also provide results in the highly-realistic *Gran Turismo Sport* race-car simulator where SSET improves learning speed and policy stability.

## 2 Terminology

Following standard definitions (Sutton & Barto, 2018), we consider a reinforcement learning agent acting in an episodic Markov Decision Process $M = \langle S, A, R, \mathcal{P}, \gamma, \mathcal{I}, \beta \rangle$ with state space $S$, action space $A$, reward function $R : S, A \to Pr[\Re]$, transition kernel $\mathcal{P} : S, A \to Pr[S]$, discount factor $\gamma \in [0, 1)$, initial state distribution $\mathcal{I} : Pr[S]$, and episode termination function $\beta : S \to \{0, 1\}$ . At time step $t$, the agent uses its current behavior policy $\pi_t : S \to Pr[A]$ to select an action and then observes the reward $r_t \sim R(s_t, a_t)$ and next state $s' \sim \mathcal{P}(s_t, a_t)$. If $\beta(s') = 1$ or a horizon of $T$ is reached, then the episode ends. The value function of a policy is defined by its long-term discounted return: $Q^\pi(s, a) = R(s, a) + \gamma E_{s' \sim \mathcal{P}(s,a)}[V^\pi(s')]$, where $V^\pi(s) = Q^\pi(s, \pi(s))$. The agent is tasked with finding an optimal policy $\pi^*$ and the corresponding $Q^*(s, a)$ that maximizes the expected discounted return. In this paper we focus on model-free off-policy methods that learn $Q^*(s, a)$ directly from data through incremental updates, such as Q-learning's (Watkins & Dayan, 1992) gradient-style update to the current value function $Q_k(s, a)$: $Q_{k+1}(s, a) = (1 - \alpha)Q_k(s, a) + \alpha\delta$ with learning rate $\alpha \in (0, 1]$ and temporal difference (TD)-error $\delta = r(s, a) + \gamma V_k(s') - Q_k(s, a)$ for $V_k(s') = \max_{a'} Q_k(s', a')$.

In deep reinforcement learning, $S$ is typically continuous and high dimensional, so the value function and (sometimes) policy are represented by neural networks with parameters $\theta_{ik}$ for each network $i$. To update the value networks, model-free deep RL algorithms like DDQN (Van Hasselt et al., 2016) make updates to $Q(s, a|\theta_k)$ along the gradient of the TD-error. To improve stability, deep RL methods typically utilize a fixed *target network* for computing $V_k(s')$ that is only updated after a batch of updates.

Experience replay (Lin, 1992) is a technique used in off-policy RL to improve sample efficiency by performing gradient updates based on many experience tuples $\langle s, a, r, s' \rangle$ stored in an ERB (see Algorithm 2 in the Appendix as an example). In the deep learning case, the ERB often stores tens or hundreds of millions of tuples with mini-batches sampled from the buffer and then used in gradient updates to $\theta_{ik}$.

Formally, we define an *event specification* (*event spec*): $\nu = \langle \omega, \tau \rangle$, composed of a (Boolean) *event condition* over states $\omega : S \to \{0, 1\}$ and a history length $\tau$. We say an event *occurs* in state $s$ if $\omega(s)$ is true. In this

paper, we assume event conditions are specified by domain experts or RL practitioners; we provide guidance on selecting a default set of useful event conditions (refer to Section 8). Section A.1 proves that event conditions that are true only in states that are visited more often by the optimal policy than the behavior collection policy (see Definitions 2, 3, and 4 in the Appendix) yield sample efficiency gains when paired with sufficiently long histories. Terminal goal states, high reward states, bottleneck states, or important rare states (e.g. passing another car) are all strong candidates. To avoid under-sampling any crucial states, histories $\tau$ must be long enough (in expectation) to reach back to a previous event occurrence, an initial state, or the horizon and chain together from $\mathcal{I}$ to the optimal policy's final state(s) (see Figure 1). Finally, outside of the core theory, when using function approximation, negatively rewarding states that are not often encountered by the optimal policy may be useful event conditions to avoid catastrophic forgetting of these possible outcomes from nearby states.

## 3 Related Work

Many approaches have been proposed for prioritized ERB sampling. The most widely used is Prioritized Experience Replay (PER) (Schaul et al., 2016), which prioritizes state/actions with the largest TD errors. However, PER does not specifically focus on states aligned with the optimal policy: indeed experiences that have zero TD error under one policy may never be sampled again even after the behavior policy has changed. In addition to empirical comparisons against PER, we show that SSET can be used with PER to leverage the benefits of both approaches: focusing leaning on high-value event trajectories aligned with the optimal policy, but also prioritizing states along those trajectories with high Bellman error. Other methods that augment vanilla PER with prioritization based on model error (Oh et al., 2021) or a meta-learning process (Zha et al., 2019) could similarly be used in conjunction with SSET .

The importance of sampling along "good" trajectories was explored in classical RL through RTDP (Barto et al., 1995) and in deep RL (Karimpanal & Bouffanais, 2018). The latter can cause unwanted data correlation in mini-batches (de Bruin et al., 2015; 2016; Huang et al., 2019). By contrast, SSET does not attempt to use data from the same trajectory in a mini-batch, instead relying on sampling to spread trajectories across many mini-batches, providing both stability and backups along a trajectory. Variants of Topological Experience Replay (Hong et al., 2021; Lee et al., 2019) also attempted to prioritize backups along a trajectory using a graph embedding originating from goal states. By contrast, SSET does not require goal states or a discrete state embedding.

Event Tables generalize the ideas explored in several "multi-table" partitioning schemes for ERBs. In (Narasimhan et al., 2015) and (Sharma et al., 2020) different tables are used to store high (or high and low) reward transitions separately from common transitions and stratified sampling is used to construct mini-batches. By contrast, SSET allows for any state-based event to partition the ERB, and more importantly stores trajectories that led to events, not just the events themselves, which is essential to ensure the sample complexity guarantees. Empirical comparisons to a reward-based event approach without histories are provided in Section 5.6. Lucid Dreaming (Du et al., 2022) stores trajectories that produce better Monte Carlo returns than the current value estimates in a separate table, but relies on generative access to the domain to create them, which is not a requirement for SSET. (de Bruin et al., 2016) keeps two buffers, one for on-policy data and another more "uniform" buffer, but requires heavy kernel computation to maintain them.

A closer comparison is the use of multi-tables in the training of a simulated race car (Wurman et al., 2022). There, the ERB is partitioned into multiple tables for different skill training scenarios based on initial states (e.g. tables for driving alone, driving in traffic, etc.) Event Tables are a generalization of this idea where data is dynamically mapped to tables rather than partitioning on an episode's initial state.

SSET also bears resemblance to multi-task and lifelong learning RL algorithms that balance the amount of data used from different tasks. Some of these approaches (e.g. Lin et al. (2019)) sacrifice general performance for success on a "main" task, or mitigate catastrophic forgetting across a sequence of tasks (Isele & Cosgun, 2018; Rolnick et al., 2019; Yin & Pan, 2017) which are not applicable in our single-task setting. Hindsight Experience Replay (Andrychowicz et al., 2017) augments the monolithic ERB in a multi-task setting by imagining goals that the trajectories could have achieved. SSET is not restricted to the multi-task setting

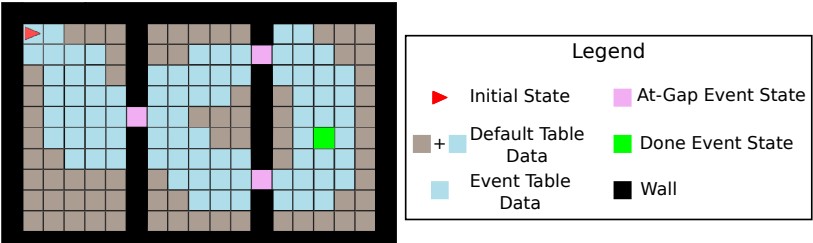

Figure 1: An example MiniGrid domain with event conditions for reaching the goal or a gap between rooms. Blue squares indicate the "fast lane" of states that can be over-sampled because they appear in both the event tables and the default table. Grey states appear only in the default table.

but could be combined with HER in such cases. CAGrad (Liu et al., 2021) constructs gradients in a way that minimizes policy degradation on the worst affected task, and is a natural pair with SSET, where updates to the model can be performed in a way that preserves performance in each of the ERB partitions (see experiments in Section 5.7).

SSET uses prior knowledge to ensure a focus on key areas of the state space and therefore has connections to initial state selection (Ivanovic et al., 2019) and potential-based reward shaping (Ng et al., 1999; Grzes, 2017). Section 5.2 shows SSET outperforming reward shaping on comparable states, but also demonstrates that the two can be used together to improve exploration (shaping) and focus value function backups (Event Tables). Because Event Tables often align with bottleneck states, they have a relationship to options (Sutton et al., 1999), but the mechanism in our work is through ER. While events in our case come from domain knowledge, future work could utilize subgoal discovery techniques (Kulkarni et al., 2016; McGovern & Barto, 2001) to identify potential events.

---

**Algorithm 1:** SSET: STRATIFIED SAMPLING FROM EVENT TABLES

---

    `//input parameters:`
1.   • $\nu_i = (\omega_i, \tau_i) \ \forall i \in [1, n]$: event conditions and their corresponding history lengths
2.   • $(\eta_i, \kappa_i, d_i) \ \forall i \in [0, n]$: sampling probabilities, capacity sizes, and minimum data requirements
3.   • $\mathbb{A}, \pi^b, T$ - Off-policy RL algorithm, behavior policy and episode length
4. $\mathcal{B}^0 \leftarrow [](\kappa_0), \mathcal{B}^{\nu_i} \leftarrow [](\kappa_i) \ \forall i \in [1, n]$,
5. **for** $episode \ k = 1$ **to** $\infty$ **do**
6.      $E \leftarrow []$   `// init episode buffer`
7.      **for** $t = 0$ **to** $T - 1$ *(or episode termination)* **do**
8.          $s \leftarrow$ current state observation
9.          execute action $a$ sampled from $\pi^b(s)$
10.          observe $r$ and next state $s'$
11.          store transition $(s, a, r, s')$ in $\mathcal{B}^0$ and $E$
12.          **for** $i = 1$ **to** $n$ **do**
             `//update event tables`
13.              **if** $\omega_i(s')$ **then**
14.                  store last $\tau_i$ transitions $[E_{t-\tau_i+1}, ..., E_t]$ in $\mathcal{B}^{\nu_i}$
15.              **end**
16.          **end**
17.      **end**
18.      $D \sim \mathcal{B}^0 \underset{i}{\cup} \mathcal{B}^{\nu_i}, \ \forall i \ s.t. \ |\mathcal{B}^{\nu_i}| \geq d_i$   `//sample & concat i.i.d. minibatches`
19.      Update weights for bias correction using $E$   `// (Lemma 2 in Appendix)`
20.      Update critic and policy networks using $\mathbb{A}$ and $D$
21. **end**

# 4 Stratified Sampling from Event Tables

We now formally define Stratified Sampling from Event Tables (SSET; Algorithm 1) and state our main theoretical result. The full proof appears in Section A.1. Given $n$ event specs $\{\nu_i | i \in [1, n]\}$, the ERB is partitioned into $n$ *event tables*, $\mathcal{B}^{\nu_i}$ and a "default" table $\mathcal{B}^0$. Each $\mathcal{B}^{\nu_i}$ holds only time steps where $\omega_i(s')$ was true or steps preceding the event occurrence in the $\tau_i$-length history. The default table $\mathcal{B}^0$ holds all time steps, including those with event occurrences. Data is inserted into each table in a FIFO manner with table capacities governed by parameter $\kappa_i$ and later used to construct training data for an off-policy RL algorithm $\mathbb{A}$.

On each agent step in SSET, experience $\langle s_t, a_t, r_t, s_{t+1} \rangle$ is stored in $\mathcal{B}^0$ as well as an ephemeral buffer $E$ for the current episode (line 11). If $\omega_i(s_{t+1})$ is true (line 13), the experience and all the $\tau_i$ steps preceding the event that were not already sent to $\mathcal{B}^{\nu_i}$ are added there as well. Intuitively, each table contains data necessary to train the value function in the area approaching an event occurrence and chain together to form a "fast lane" (Figure 1) for backups that will be over-sampled compared to monolithic ER. For simplicity Algorithm 1 assumes each event spec maps to a unique table (lines 4 and 14) but the mapping could also be surjective. Note that the (potentially overlapping) data stored in these tables can be managed efficiently by ERB implementations like Reverb (Cassirer et al., 2021).

When mini-batches are constructed (line 18), fixed i.i.d. proportions are collected from each table using probabilities $\eta_i$, for $i \in [0, n]$. To prevent early over-fitting, minimal data requirements (based on the mini-batch size) can be applied, with proportions normalized appropriately if one or more tables has insufficient data.

Like PER and other ERB prioritization schemes, SSET can introduce bias in stochastic environments. In the extreme case, if $\mathcal{P}(s, a)$ has equal probability for two terminal states $s_1$ and $s_2$ but only $\omega(s_1)$ is true, then sampling from the event table will skew the data distribution higher for $s_1$ outcomes. To alleviate this error, line 19 applies a weighted correction derived in Lemma 2 (Appendix). The full correction term applies a weight based on the probability of $\langle s, a \rangle$ *not* being in each event table, which can be computed in discrete domains by counting the number of times a $(s, a)$ pair was not part of any event history. For continuous domains, a priority sum tree data-structure could be used similarly to PER (Schaul et al., 2016) by setting priorities for samples in the event buffer equal to $(1 + \eta)$ and samples outside to $(1 - \eta)$. The correction weights for each transition inside the event tables would be $\frac{\text{priority-sum}}{1+\eta}$. However, this approach is aggressive and might slow down learning, as seen in PER. Note, for deterministic MDPs the correction is not needed. In environments without non-goal terminal states, increasing $\tau$ may mitigate some bias by storing more non-event outcomes in the event table. In our mildly stochastic empirical studies we use this longer $\tau$ approach.

We now state the main theoretical result of the paper, with the full proof in Section A.1. Intuitively, the theorem states that SSET using event conditions that are correlated with an optimal policy and histories that are sufficiently long, will have sample complexity $N^{\mathcal{B}^\nu \cup \mathcal{B}^0, K}$ for learning $Q^*(s, a)$ in the necessary part of the state space ($\mathcal{S}^f$), and $N^{\mathcal{B}^\nu \cup \mathcal{B}^0, K}$ is (with high probability) smaller than the sample complexity of learning with a monolithic buffer of the same size.

**Theorem 1.** *Let $\mathcal{S}^f = \{s \mid P(\Gamma_{s,\cdot}^{\pi^*} \subset \mathcal{B}^\nu) \geq \bar{p}\}$ denote the set of states s.t. the sampled optimal trajectories ($\Gamma^{\pi^*}$; Def. 2) starting from those states are contained in the combined event-buffer with a probability greater than $\bar{p} \in (0, 1]$, and $\eta = \sum_{i=1}^n \eta_i$. Under the conditions of Prop. 1 and using $\mu$ as defined in Lemma 1 if $\tau_{\forall i \in [1,n]} \leq \frac{(1-\eta)^m}{(m+1)n\mu}$, then at iteration $K$ of Q-learning with a target function:*

$$P\left(N^{\mathcal{B}^\nu \cup \mathcal{B}^0, K} \leq (1-\eta)^{2m} N^{\mathcal{B}, K}\right) \geq \bar{p}, \forall s \in \mathcal{S}^f, m \in \mathbb{Z}^{0+}$$

*Proof sketch.* We make use of lower bounds (Li et al., 2022) on the convergence rate of *tabular* Q-learning with a target function (Algorithm 2 in the Appendix). We define the state probability distribution (density) following a given policy to a finite horizon from an initial state and use that to define the state density disparity to the optimal policy. We then formally define *event conditions* on the states with low optimal-policy disparity or final states of the optimal policy. We extend this definition to event sections and their

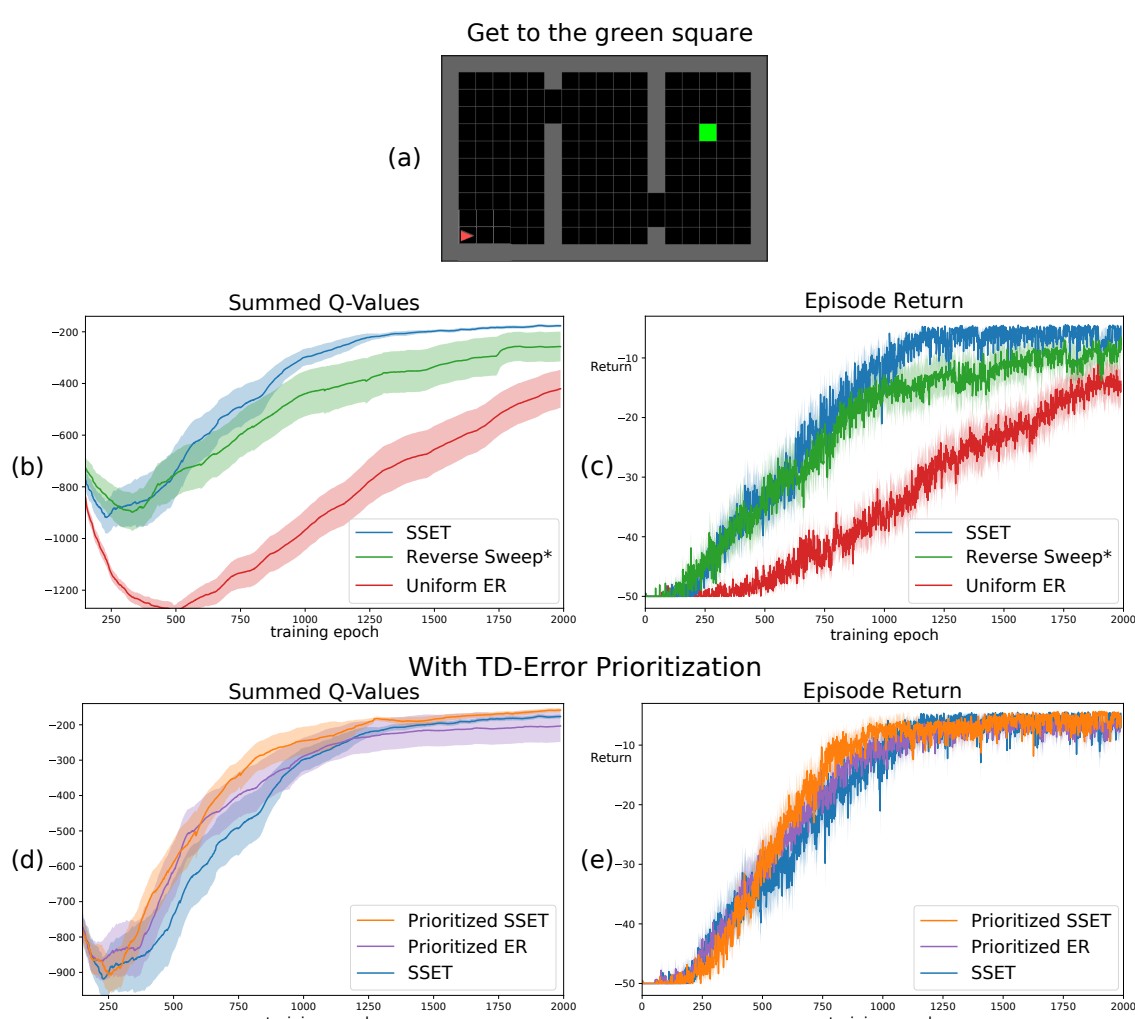

Figure 2: **Three Room Grid World** Statistical results from 30 randomly seeded runs (mean shown using solid lines and std-error using shaded regions) (a) Environment (b) Learned target Q-values summed across the entire state-action space vs training epoch. SSET Q-values converge with a significantly low std-error. (c) Episode return vs training epoch (d)-(e) Results with TD-error prioritized sampling.

corresponding tables that include sufficient history to (on expectation) reach back to a previous event or initial state. We then quantify (Lemma 1) the over-sampling of experience in the event tables and derive the convergence rate (Prop. 3) and bias correction procedure (Lemma 2). Finally, we show that the resulting convergence bound is an improvement over uniform sampling. □

## 5  MiniGrid Experiments

We now provide experimental validation of SSET in environments from the MiniGrid (Chevalier-Boisvert et al., 2018) domain using the DDQN (Van Hasselt et al., 2016) RL algorithm. We use a dense neural net architecture with hidden layers to encode the Q-functions and an $\epsilon$-greedy behavior policy for collecting data. In each experiment SSET demonstrates improvements in sample complexity or decreased variance against uniform experience replay (Uniform ER), an off-policy version of an (on-policy) reverse sweep of updates from the goal (Reverse-sweep*; in the spirit of EBU (Lee et al., 2019)), and TD-error prioritized experience replay (PER) (Schaul et al., 2016).

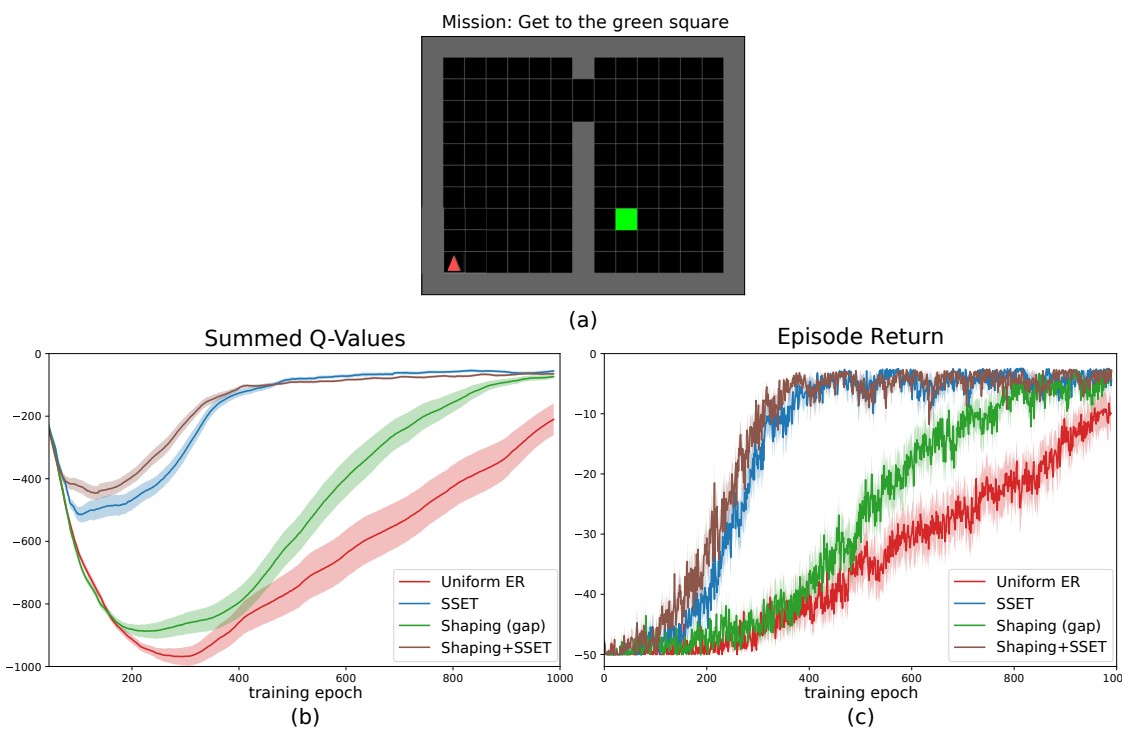

Figure 3: **Comparison with Intermediate Shaping Rewards**. Results from 30 randomly seeded runs comparing SSET against using intermediate shaping reward at the gap. (a) Environment (b) Learned target Q-values summed across the entire state-action space vs training epoch. (c) Episode return vs training epoch.

## 5.1 Proof of Concept: Three Room Grid World

We first demonstrate the sample complexity speedup of SSET in a three-room world (Figure 2(a)). The agent's observation is its 3D grid position and orientation $(x, y, \theta)$ and its available actions are *turn-left*, *turn-right* and *forward*. The rewards are $+1$ at the green square and $-0.1$ otherwise. We use two event conditions for SSET with a history length of 200: one that occurs at any gap state between two rooms, and another that occurs at the green square. Refer to the Appendix for more details. Figures 2(b)-(c) show the mean and standard error (from 30 randomly initialized runs) of the learned Q-values summed across the entire state-action space and the resulting episodic return during the course of training. SSET performs best in terms of both sample efficiency and learning stability (lower variance between runs). The result against a reverse-sweep approach demonstrates the utility of the intermediate gap events, which serve as waypoints for the backups. Figure 2(d)-(e) compares SSET to PER and a combination of the two. PER speeds up learning compared to uniform sampling, but with considerably more variance across seeds than SSET, possibly due to overestimation bias. Combining SSET with TD-error prioritization yields the best of both worlds; providing reliable local Bellman target estimates and prioritizing samples with high error to those targets.

## 5.2 Comparison with Shaping Rewards

Potential-based shaping (PBS) (Ng et al., 1999; Grzes, 2017) provides a framework for adding rewards to speed up learning without affecting the optimal policy. Similar to PBS, SSET preserves optimality (see Lemma 2 in Appendix) while providing an effective way to infuse domain knowledge. Here, we compare the performance of SSET and reward shaping as well as their combination in a grid-world domain similar to the previous experiment (Figure 3(a)). A shaping potential with $\phi(s) = 1$ at the gap and 0 elsewhere is used to provide a reward component of $(\gamma\phi(s') - \phi(s))$ in addition to the environmental (goal) reward. SSET uses events that occur at the gap and the goal. Figures 3(b)-(c) show SSET outperforms both shaping and

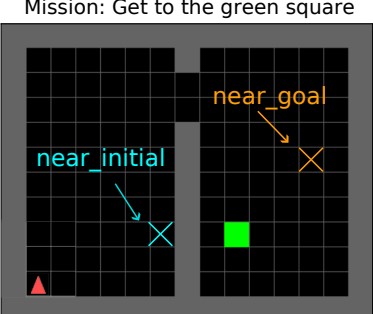

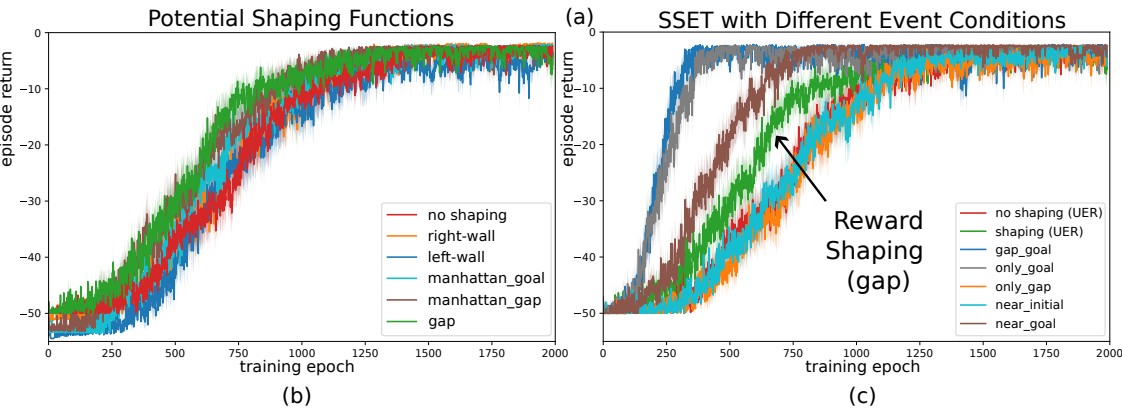

Figure 4: **SSET performance with ideal and less-ideal event-conditions** (a) Environment (b) Comparisons of different potential-based reward shaping functions with uniform ER to use as baselines (c) SSET's performance of ideal and less-ideal event conditions along with baselines (Uniform ER with our best shaping function and with no shaping rewards).

the no-shaping baseline. While reward shaping acts as a heuristic to speed up exploration, those rewards still need to be bootstrapped back to the initial states, which is the forte of SSET. Thus, when the two are combined (brown line), even better performance is realized as shaping guides exploration and SSET provides a "fast lane" to bootstrap the values. Comparisons to less ideal potential shaping functions and ablations of the $\eta_0$ parameter for SSET in this domain are provided in Sec. 5.3.

### 5.3   SSET Performance with Badly Designed Event Conditions

Based on the formal definition of event conditions (Def. 4), good events occur in states that are aligned along optimal trajectories. We assume that such event conditions are typically specified by domain experts or RL practitioners. In this section, we evaluate the performance of SSET when the event conditions are badly designed and compare them against good conditions, along with other baselines such as uniform ER with different potential reward-shaping functions. Finally, we also study how performance varies with changes to the default buffer's sampling probability ($\eta_0$).

Figure 4(a) shows results from the 2D grid world domain used in the reward shaping comparisons (Section 5.2). The task is to get to the green square starting from an initial bottom-left grid position, which requires navigating through the gap between the rooms. We first compare several potential reward-shaping functions to pick a good baseline. Figure 4(b) shows the statistical means (computed from 30 randomly seeded runs) of episodic-return during training using different potential shaping functions: gap (+1 at the gap grid position and zero everywhere), manhattan_goal (normalized manhattan distance between agent's position and the goal), manhattan_gap (distance to the gap), right-wall (normalized x-distance to the right boundary wall), left-wall (normalized x-distance to the left boundary wall) and no-shaping. Not surprisingly, the best performing shaping function is the "gap" function, which motivates the agent to get to the gap first. Others perform either similar or subpar to the experiment with no-shaping rewards.

# Performance vs Default Buffer Sampling Weight ($\eta_0$)

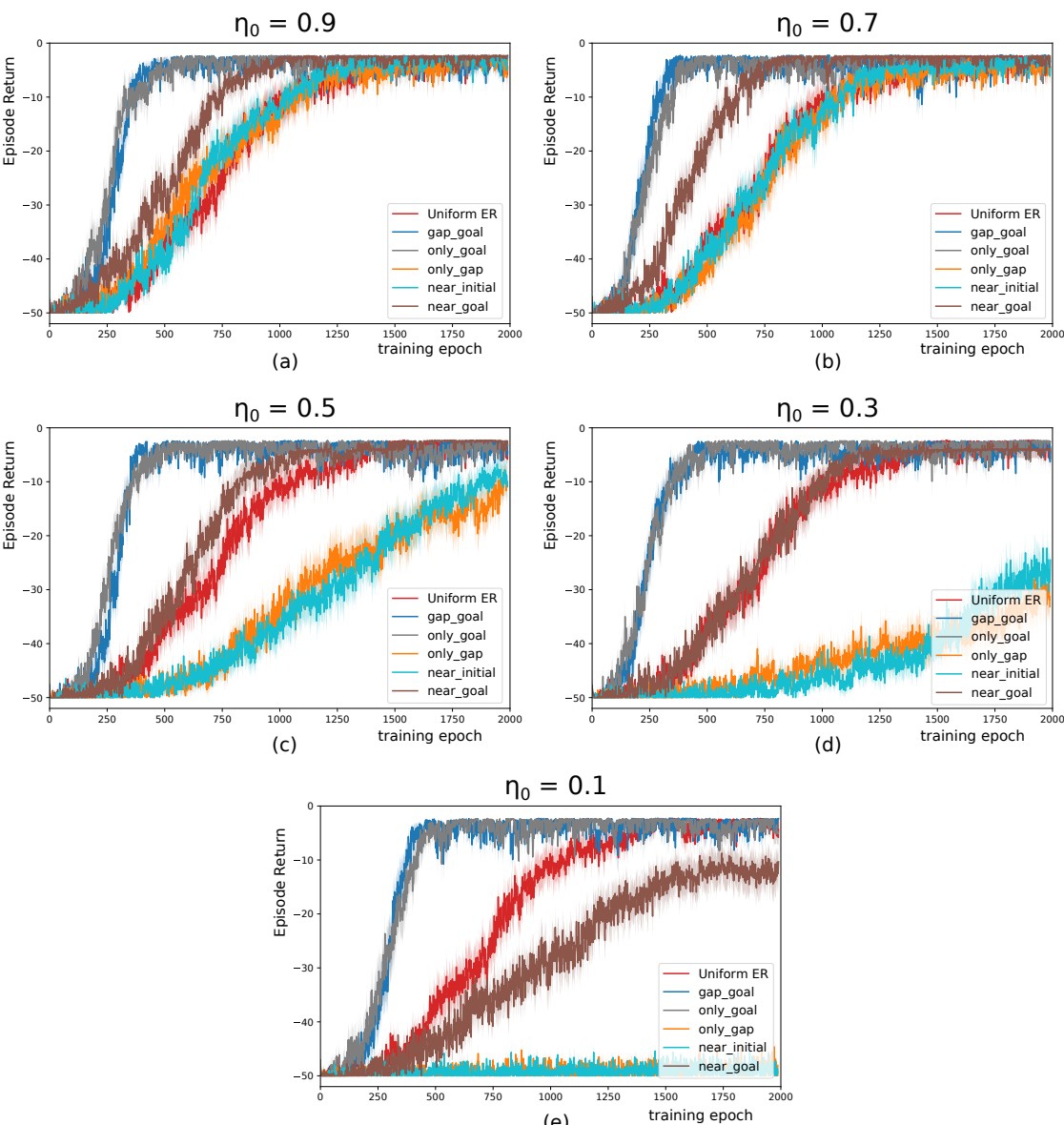

Figure 5: **SSET performance vs default-buffer sampling weight** Statistical means of episodic-return from 30 randomly seeded runs for different values of $\eta_0$ and different event-conditions. Experiments with good conditions (gap_goal, only_goal) are least affected by the changes in $\eta_0$. The experiment with the mediocre condition (near_goal) seems to benefit from fine-tuning (best at 0.7), and the experiments with bad conditions (near_initial, only_gap) suffer due to under-sampling the default buffer.

Next, we compare SSET performance for different event conditions and $\eta_0 = 0.7$: two conditions (based on Def. 4) that trigger at the goal and gap (gap_goal), or goal alone (only_goal) with a sufficient history length ($\tau = 30$). We selected three less-ideal conditions that occur at positions near the goal (near_goal), at the gap (only_gap) and near the initial position (near_initial), respectively. Figure 4(c) shows statistical means of the corresponding experiments along with the baselines of shaping (using the gap potential function) and uniform ER with no shaping. The plot shows that there is a significant improvement in learning efficiency for experiments using the well defined events. The performance of SSET of near_goal event condition is not

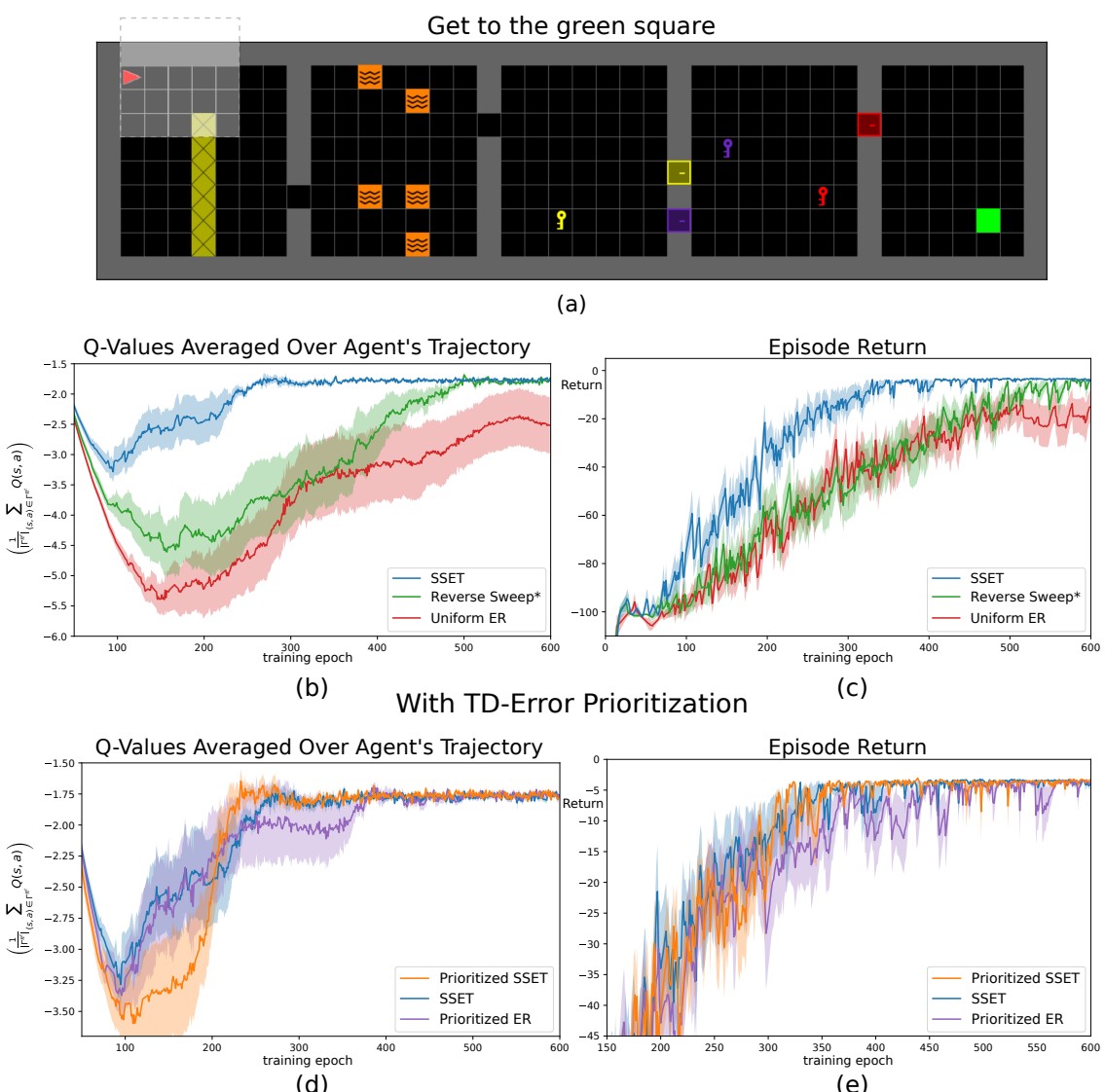

Figure 6: **Obstacle Course** Results from 30 randomly seeded runs (mean shown using solid lines and std-error using shaded regions) (a) An environment with spikes (yellow), lava (orange), colored keys and locked doors (b) Learned target Q-values averaged over agent trajectories vs training epoch (c) Episodic return vs training epoch (d)-(e) Results with TD-error Prioritized Sampling

ideal, but still much better than our best shaping baseline. Experiments with only_gap and near_initial conditions perform similar to the Uniform ER (no-shaping) as learning relies significantly on transitions in the default buffer to bootstrap the value from the rewarding states (goal reward in this case) to the states where the events occur. Based on these results, we conclude that both reward shaping and event tables show degradation when using poorly chosen shaping functions or event conditions but that even with poorly chosen events SSET often outperforms reward shaping in this domain.

Next, we compare how SSET's performance varies with default-buffer's sampling probability (Figure 5). The plots show that experiments with good conditions are least affected by the changes in $\eta_0$. The experiment with the mediocre condition (near_goal) seems to benefit from fine-tuning (best at 0.7), and the experiments with bad conditions suffer ($\eta_0 < 0.7$) due to under-sampling the default buffer. For all practical purposes, we recommend using higher values for $\eta_0$ unless the event conditions are carefully designed.

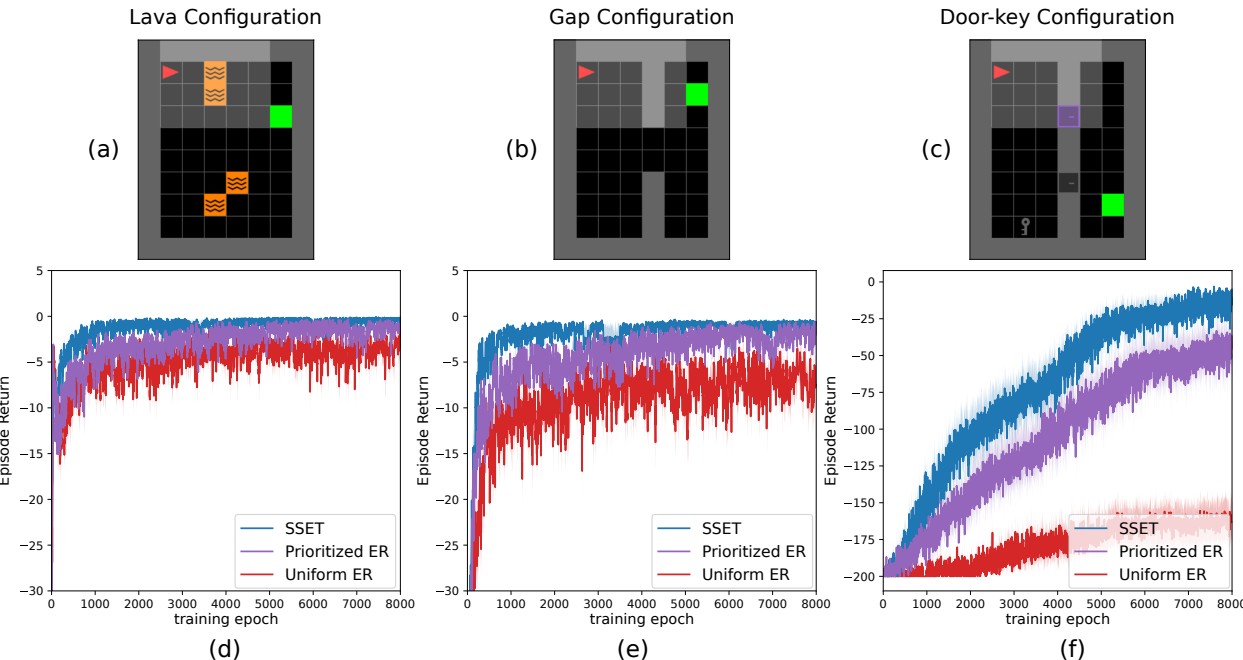

Figure 7: **Randomized Skill Environment** Statistical results from 30 randomly seeded runs (mean shown using solid lines and std-error using shaded regions) (a)-(c) Instances of randomly sampled object configurations. (d)-(e) Episodic return for each skill during the course of training.

## 5.4 Obstacle Course and Randomized Skill Environment

The more complex *obstacle course* environment consists of multiple sections (Figure 6(a)) containing spikes with -1 reward (yellow squares), lava with -1 reward and episode termination (orange squares), and colored keys to open doors. The exact positions of lava, gaps in the walls, keys, doors, and key/door colors are randomly set in each episode but the ordering of the rooms is fixed (e.g. the spikes are always in the first room). The agent's task is to get to the green square starting from the top-left corner. The agent's observation consists of (a) an egocentric $5 \times 5$ localized forward-view image (highlighted region in Figure 6(a)) that encodes a representation of obstacles around it, (b) a Boolean indicator for carrying an object, (c) a 2D representation (category, color) of the object it is carrying, otherwise $(-1, -1)$, and (d) 3D grid position and orientation $(x, y, \theta)$. The agent's action space includes 3 additional actions (*pickup key, drop key, toggle door*). We use event conditions with a history length of 50 associated with each obstacle category (e.g. picked-up-a-key, opened-a-door, etc. see Appendix for more details). Figure 6(b)-(c) clearly illustrates the improved efficiency and stability of using event-tables over reverse-sweep and uniform ER. In comparison to TD-error prioritization (Figure 6(d)-(e)), again PER exhibits more variance across multiple seeded runs. Interestingly, in this domain, SSET performs equally well with or without additional TD-error prioritization.

Finally, we consider a randomized multi-skill setup where the agent must either avoid lava, go through a gap, or open the correct door to reach the goal (Figure 7(a)-(c)). The scenario (lava, gap, or open-door) and object positions / colors are sampled randomly for each training episode. This is a more challenging task than the obstacle course above because object locations don't follow a fixed sequential pattern. A square that contains a door in one episode may contain lava in the next one. Using uniform ER, value function learning is dominated by the easier skills (lava, gap) and the agent fails to acquire the difficult open-door skill. Learning using SSET performs much better, acquiring and maintaining all the skills compared to uniform sampling and even PER.

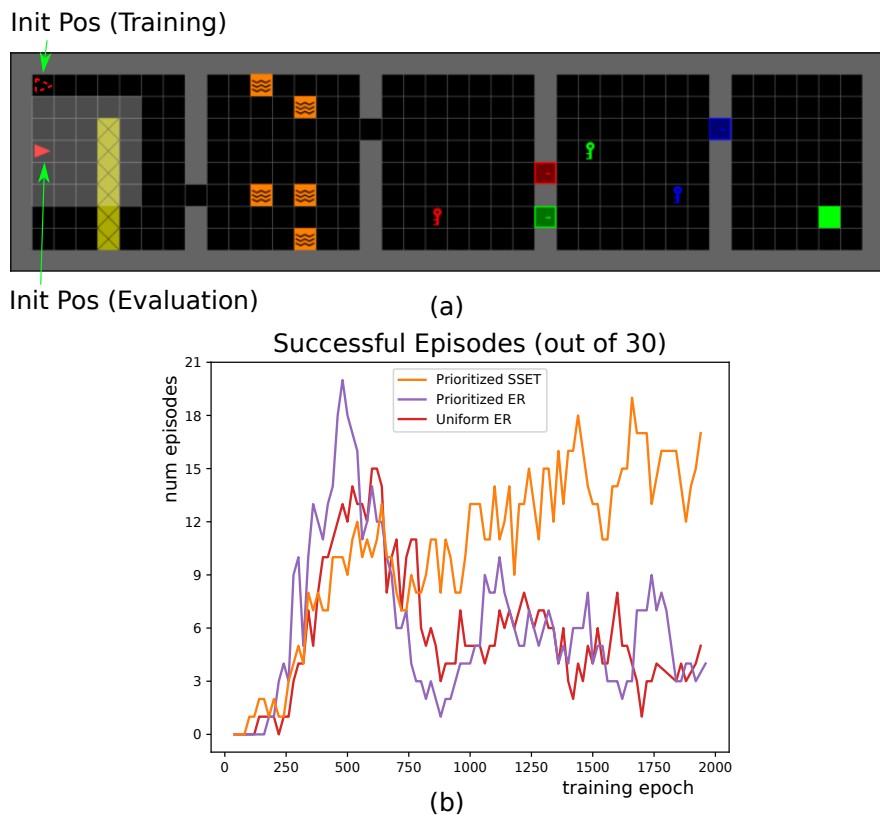

Figure 8: **Evaluation Result of a Slightly Shifted Initial Position** Test-time performance of the agent starting from a slightly shifted initial grid position $(1, 4)$. (a) Environment instance (b) Number of successful episodes out of 30 seeded runs of the experiment for each evaluation checkpoint.

## 5.5 Catastrophic Forgetting in Obstacle Course

Here, we illustrate how SSET avoids catastrophic forgetting that hampers even PER during extended learning in the obstacle course experiment presented in Section 5.4.

At the end of each episode during training, the agent is always reset to the top-left corner $(1, 1)$ of the environment. At a frequency of every 20 epochs during training, we evaluated the performance of the agent starting from a slightly shifted initial grid position $(1, 4)$ as shown in Figure 8(a). Figure 8(b) shows the number of successful episodes over the 30 seeded runs of the experiment for each evaluation checkpoint. An episode is considered successful if the agent is able to navigate through the obstacles and reach the green square. The plot shows that the agents with both uniform and TD-error prioritized experience replay learn how to complete the task early on, but forget the skill as training continues. We hypothesize that due to changing epsilon-greedy behavior policy, the Q-function updating with samples from a standard FIFO uniform (and even prioritized) replay buffer, quickly begins to over-fit trajectories starting from $(1, 1)$ avoiding the nearby negative rewarding spikes. Over-fitting on these trajectories potentially leads to losing previously learned estimates for nearby transitions. On the other hand, with SSET and sampling from the *at-spike* event table that is unaffected by the updating behavior policy, the agent continues to improve its value estimates for those transitions over time resulting in a better behavior.

In the context of this particular experiment and similar RL domains, having robust performance against perturbations is not typically expected and therefore catastrophic forgetting may not be a serious issue. However, it becomes a challenging issue to tackle in realistic domains like Gran Turismo where the training and testing distributions can be different or the domain is simply so large that one can forget low probability events, like leaving the course in Section 7.2.

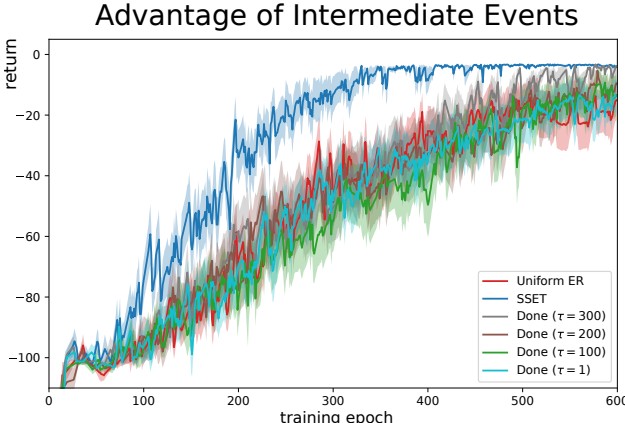

Figure 9: Episodic return from 30 randomly seeded runs (mean shown using solid lines and std-error using shaded regions) in the obstacle course environment comparing SSET with intermediate events like, pickup-key, at-door, etc. against only using a done-conditioned event with different history lengths.

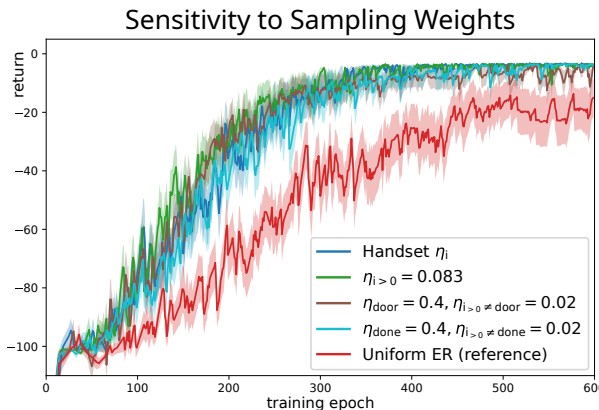

Figure 10: Episodic return from 30 randomly seeded runs (mean shown using solid lines and std-error using shaded regions) in the obstacle course environment comparing different sampling weights $\eta_i$ keeping the default table's sampling weight fixed $\eta_0 = 0.5$.

## 5.6 Intermediate Events, Histories, and Sampling Weights

The proof of Theorem 1 shows that SSET improves sample complexity along transitions from the optimal trajectory that are, with high probability, stored in the Event Tables. We can increase this probability by either increasing the number of event conditioned tables $n$ with short histories or having fewer tables with longer histories $\tau_i$. However, the proof of Lemma 1 shows the latter is not as effective, especially in early learning when the histories leading to a far-off event may contain many sub-optimal actions, diluting the impact of that table until the policy is better optimized. Instead, when possible, it is best to have many events acting as waypoints in the environment as shown in Figure 1. We now present an empirical test supporting this insight in the sparse-reward obstacle course environment presented in Section 5.4. The test doubles as a comparison to methods that partition an ERB based solely on reward conditions without corresponding histories (Sharma et al., 2020).

We compare SSET with the intermediate events used in Section 5.4 (pickup-key, at-door, etc.), each with a history length of 50 (see Table 1 for parameters) against only using a terminal rewarding event with different history lengths. Figure 9 shows the statistical average curves of episodic return during the course of training computed from 30 random seeded runs. It is clear from the result that SSET with intermediate events outperforms the rest. Intuitively, intermediate events can be seen as waypoints on the fast lane for

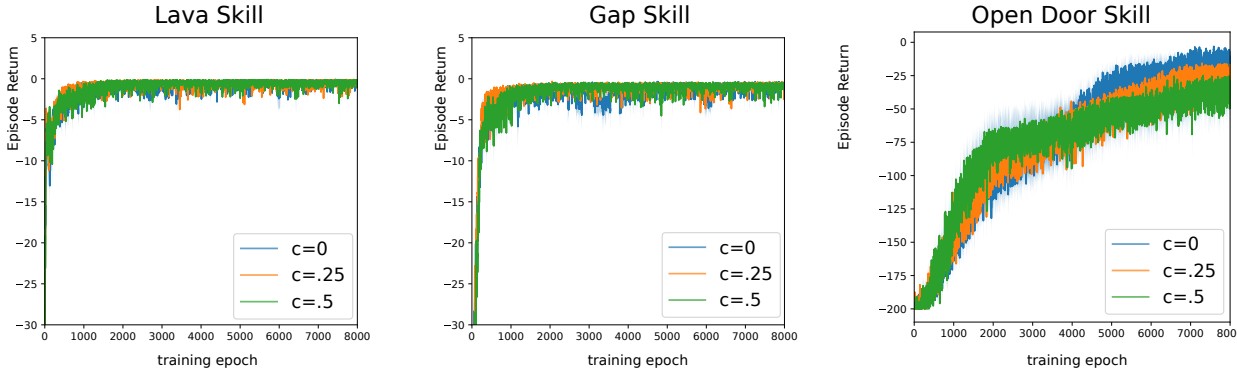

Figure 11: **CAGrad with SSET** Statistical results for different values of CAGrad's coefficient of regularization $c \in [0, 1)$ from 30 randomly seeded runs (mean shown using solid lines and std-error using shaded regions). Higher values of $c$ boost early performance but result in a lower asymptote.

TD backups to the initial states, thereby improving the overall sample efficiency. The plot also highlights that just sampling terminal goal states (as in SER (Sharma et al., 2020), $\tau = 1$ in the chart) or transition sequences leading to them (as in TER (Hong et al., 2021), EBU (Lee et al., 2019) or Reverse-Sweep*, higher $\tau$ values) may not be sufficient to achieve improved efficiency in a sparse reward setting like the Obstacle Course.

Finally, in Figure 10 we compare SSET performance for different sampling weights $\eta_i$ keeping the default table's sampling weight fixed ($\eta_0 = 0.5$). Handset $\eta_i$ are the same as the ones mentioned in Table 1 in Appendix, $\eta_{i>0} = 0.5/6 = 0.083$ are weights distributed equally across the six event tables, $\eta_{\text{door}} = 0.4$ assigns a large weight to the door event table and $\eta_{\text{done}} = 0.4$ assigns a large weight to the done table. The sampling is done such that there is at least one sample picked from each table. The results show that the performance is similar with the extreme sets showing some minor fluctuations at the end, but still performing better compared to using uniform experience replay. Our general recommendation is to start with using equal weights across all events tables and if needed tuning them further to get a peak performance.

## 5.7 Conflict Averse Gradient Descent with SSET

Most temporal-difference RL algorithms like Q-learning rely on minimizing Bellman errors to local target estimates ($Q_k$; see Alg. 2). With an increasing number of outer iterations, these local targets inch closer to the true global target $Q^*$, thereby optimizing the RL objective asymptotically. During early training, stratified mini-batch sampling from event tables could result in local gradients that may conflict with each other. Using a standard average gradient descent approach, like SGD, might not be beneficial for uncommon or difficult-to-learn events as the average gradient would be skewed in the direction of the easier events. For example, in the randomized multi-skill experiment presented in Section 5.4 (Figure 7), the agent takes longer to learn the open-door skill compared to the others. Previous multi-task learning work has proposed several approaches mitigating this problem, albeit in a multi-task pareto-optimal setting. We explore using one such recently proposed multi-objective optimization algorithm called Conflict-Averse Gradient descent (CAGrad) (Liu et al., 2021) to see if we could boost up learning the difficult open-door skill.

Figure 11 shows statistical results using CAGrad with SSET for different values for the algorithm hyperparameter $c \in (0, 1)$, which controls the extent of minimizing conflicts between losses within a local ball centered around the average gradient. Therefore, setting $c = 0$ reduces to the standard average gradient descent. The plots illustrate that higher values of $c$ boost initial learning but result in lowering the asymptotic value. This can be explained by the fact that, as the local value function targets $Q_k$ move closer to the global target $Q^*$, the conflicts between event-section losses minimize making conflict optimization redundant. We hypothesize that scheduling the parameter $c$ from high values to zero during the course of training would improve the sample complexity and reach optimal asymptotic behavior. Exploring different schedules is outside the scope of this paper and we leave this for future work.

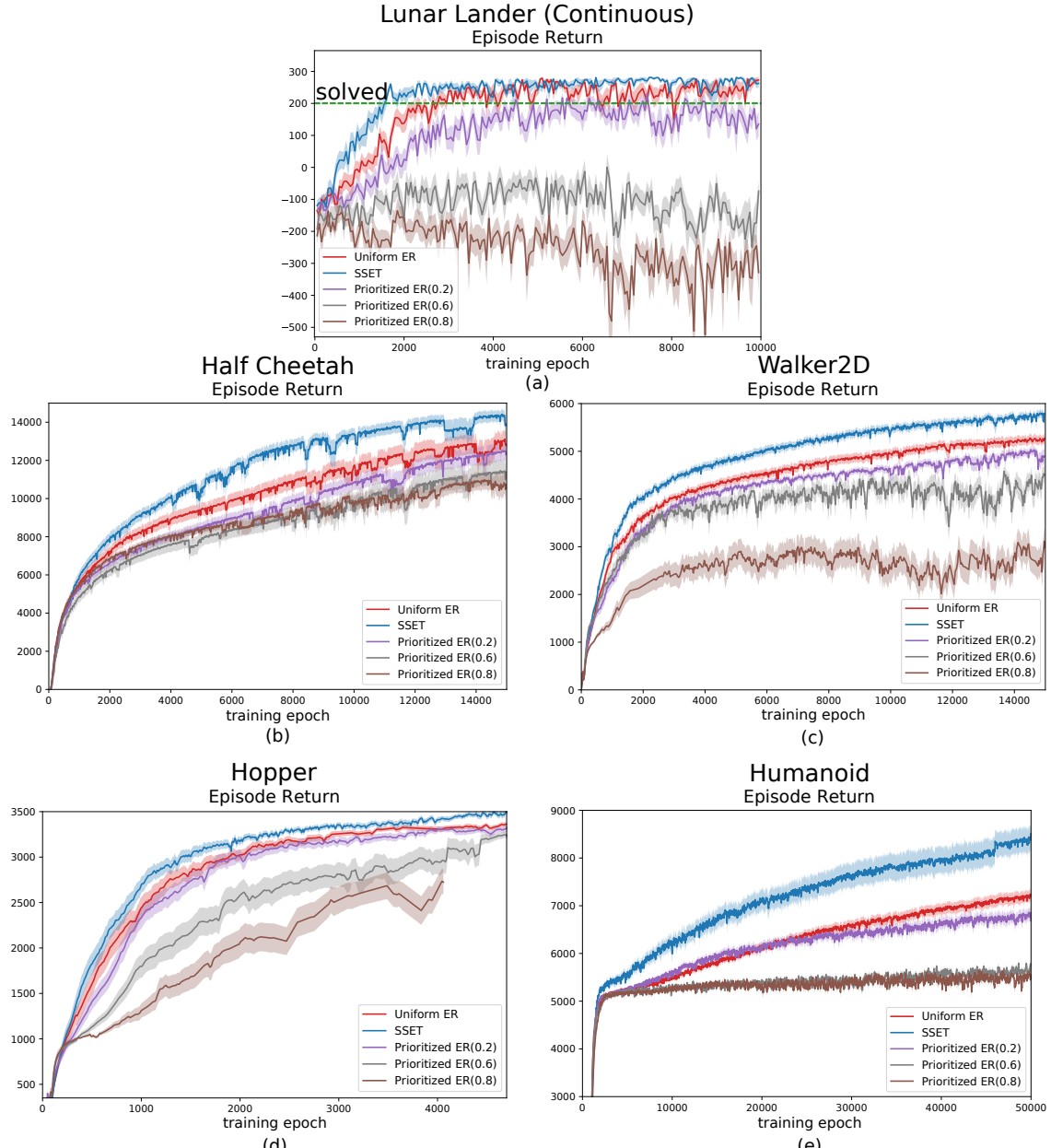

Figure 12: **Continuous Control Benchmark Tasks** SSET outperforms various parameterizations of PER and Uniform sampling in common RL benchmarks.

## 6 Lunar Lander and Mujoco Experiments

We now demonstrate the improved sample complexity and stability of SSET on several continuous control benchmark tasks (LunarLanderContinuous-v3 and MuJoCo (Todorov et al., 2012) suite defined in OpenAI Gym (Brockman et al., 2016)) that have dense shaping rewards and, in the case of the MuJoCo domains, no pre-defined goal states. For the LunarLander environment, we used two event conditions for SSET with a history length of 200: one that occurs when both lander's legs make contact between the flags, and another when the lander's position is close to the middle of the flags. For the MuJoCo suite, we used three event conditions that occur when the agent receives rewards greater than certain thresholds (See Appendix Table 2). Each of those events used history lengths of 200. The thresholds were manually selected for each environment

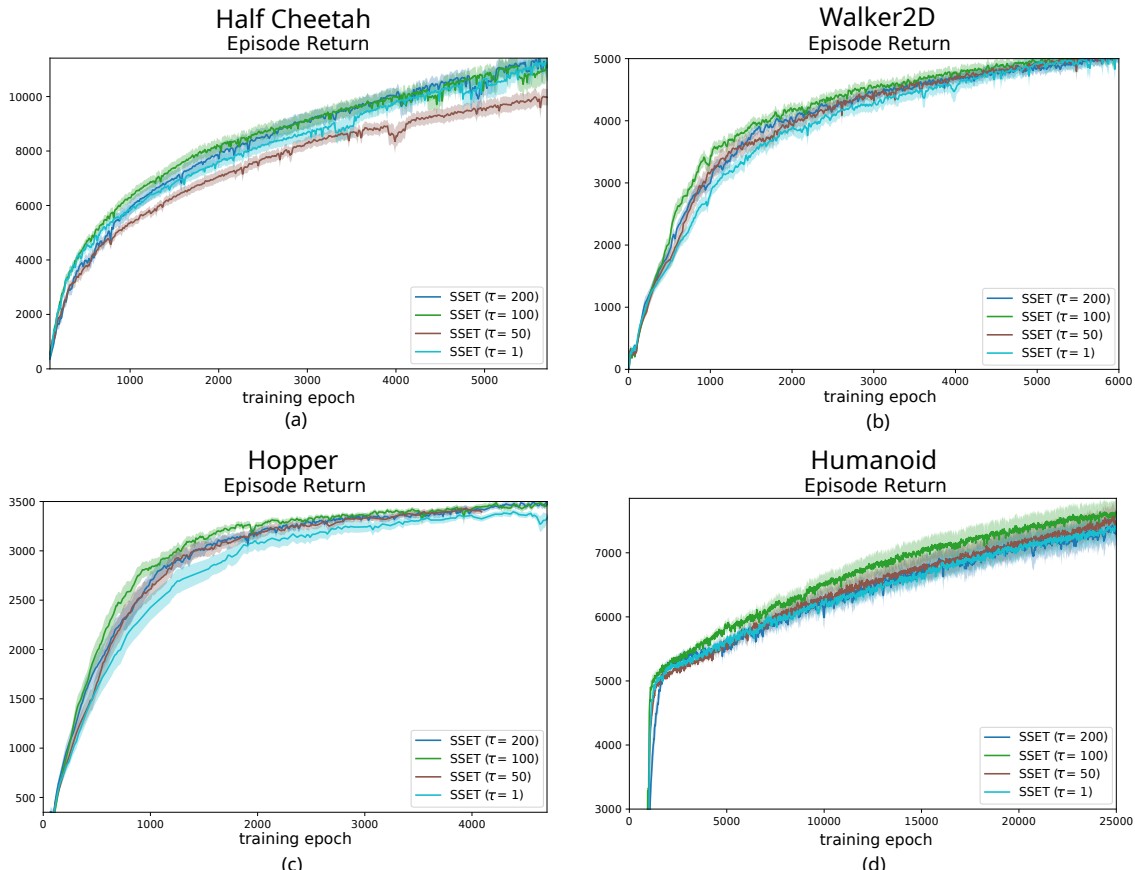

Figure 13: **Continuous Control History Lengths** ($\tau$) SSET performance in MuJoCo domains at different history lengths. SSET benefits from using sufficiently longer history lengths, which can be tuned for a peak performance.

based on the reward bounds. We used the state-of-the-art SAC (Haarnoja et al., 2018) RL algorithm to compare SSET against uniform experience replay and PER at different priority exponents. Figure 12 shows the statistical mean and standard errors of empirical returns computed from 30 randomly seeded episodes evaluated at different epochs during training. The results definitively illustrate SSET improves sample-efficiency (by roughly half the number of epochs) and achieves stable policies by bootstraping the salient rewards more rapidly. All four of the MuJoCo domains and LunarLander show similar patterns for SSET. PER performs at-best similar to the uniform experience replay and the performance degrades as we increase the priority exponent. We suspect this is because of the high density of shaping rewards, which makes the TD-Errors volatile, making them a bad match for PER.

Next we show results from an ablation study of changing history lengths in the dense reward MuJoCo domains, similar to the experiments in sparse reward domains in Section 5.6. Figure 13 shows the average episodic return (with std-error shown using the shaded regions) during the course of training computed from 30 random seeded runs. In MuJoCo domains, rewards are proportional to the continuous progress made by the agent and with events based on reward thresholds, sample histories for a higher threshold get stored in the tables corresponding to the previous threshold. The results of this study show SSET is again robust to 'non-optimal' history lengths, but still show the benefits from using sufficiently long history lengths. These results reinforce the use of longer history lengths in the absence of prior knowledge, and also show the history lengths can be tuned for a peak performance.

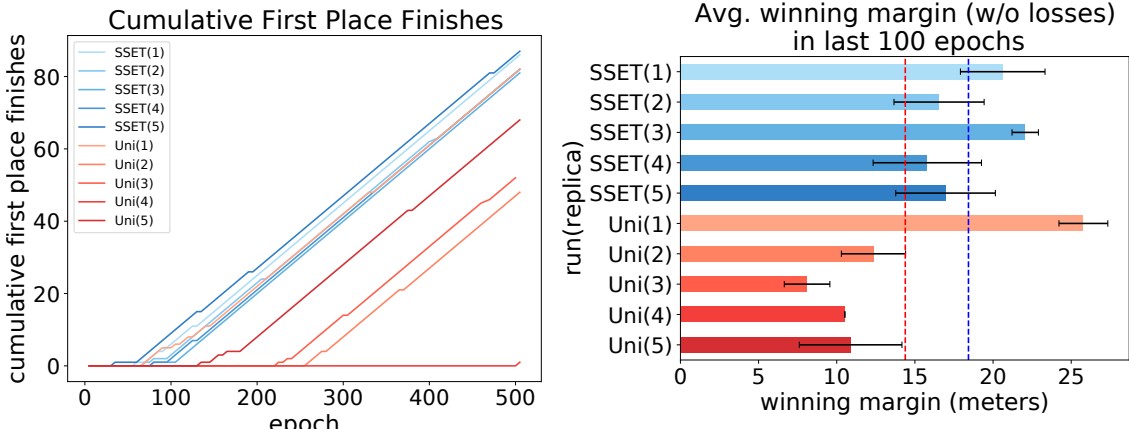

Figure 14: **Left:** Cumulative wins evaluated every 5 epochs in the slingshot passing test. Uniform sampling shows high variance while SSET with a slipstream (0.7) event and a "won" event consistently has sample complexity on par with the best uniform-sampling run. **Right:** Average (and std dev) of winning margins (excluding losses) in the last 100 epochs for each run. While one uniform ERB run did best, on average (dashed lines) SSET has the consistently better performance.

## 7 Simulated Car Racing Experiments

Gran Turismo™ Sport is a PlayStation™ racing simulator that has been previously used as an RL testbed (Fuchs et al., 2021; Song et al., 2021) and where an RL system recently outraced human e-sports champions (Wurman et al., 2022). The latter work used a multi-table ERB based on different initial state conditions, which in the experiments below is equivalent to the uniform sampling approach. We show the SSET speeds up convergence when learning to pass another car and mitigates off-course driving in a time-trial scenario.

The environment, features, and training details (see Section A.3) are the same as Wurman et al.'s except that we focus on smaller scenarios to isolate the specific effects of Event Tables. All experiments collected data at 10Hz from 21 PlayStations with one of those typically dedicated to evaluation tasks. The state representation includes hundreds of state features covering aspects such as 3-D velocity, steering angle, a representation of the upcoming course points, and a representation of opponent cars including a $[0, 1]$ measure of the *slipstream* produced by a car ahead. The Quantile Regression SAC (QR-SAC) algorithm is used with $2048 \times 4$ feed-forward neural networks for the value functions and policy. Dropout is used when training the policy network.

### 7.1 Learning the "Slingshot" Pass

In the first experiment, we demonstrate SSET 's sample complexity benefits in a "slingshot passing" scenario on the Circuit de la Sarthe (Sarthe) track, using a Red Bull X2019 Competition race car, similar to a Formula 1 vehicle. The environment is a relatively straight 1700 meter section of the course with the RL agent always launched behind (in training between $[10, 40]$ meters) one built-in-AI from the game. To succeed, the agent needs to use the opponent's slipstream to accelerate beyond its top open-air speed and use the added momentum to slingshot by the opponent and hold it off to the end of the section. Reward function components incentivize course progress and passing and penalize wall hits, car collisions and off-course driving. Full details are provided in the supplemental material.

For this task, we introduce two events. A "slipstream" event (with $\tau = 10$s) occurs when the agent's slipstream feature is above a threshold (0.7 in this case) and a "won" event (with $\tau = 15$s) occurs if the agent ends the section in first place. Both events use $\kappa = \eta = 10\%$. Figure 14 reports the cumulative wins for 5 replicas each of SSET versus a monolithic ERB with uniform sampling, both with total capacities ($\sum \kappa_i$) of 2.5 million steps and sharing common seeds. Policies were evaluated after every 5 epochs with the agent started 35 meters behind the opponent. The uniform sampling runs display high variance, taking anywhere

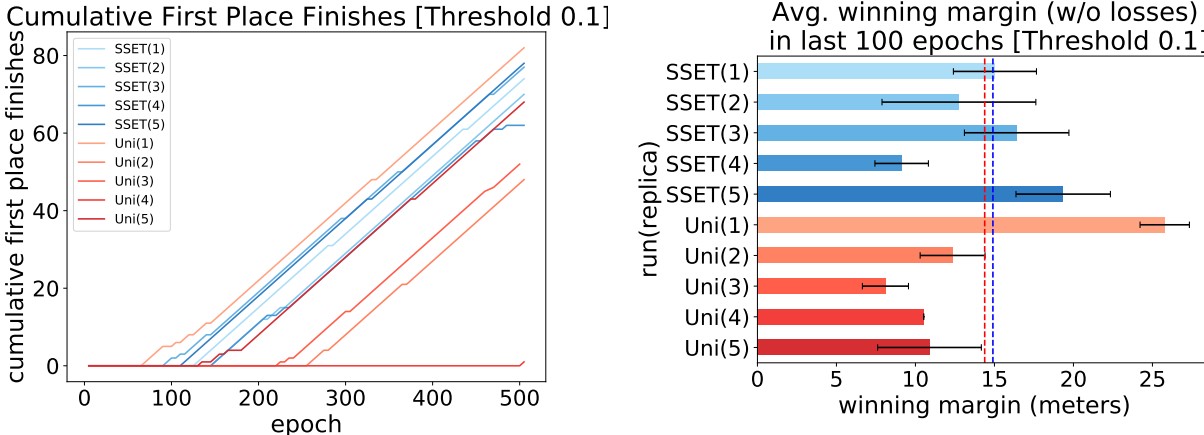

Figure 15: **Left:** Cumulative wins evaluated every 5 epochs in the slingshot passing test with a threshold of **0.1** (almost any slipstream effect). Uniform sampling shows high variance while SSET with a slipstream (0.1) event and a "won" event has better variance but overall performance is not as good as the results with a 0.7 threshold. **Right:** Average (and std dev) of winning margins (excluding losses) in the last 100 epochs for each run.

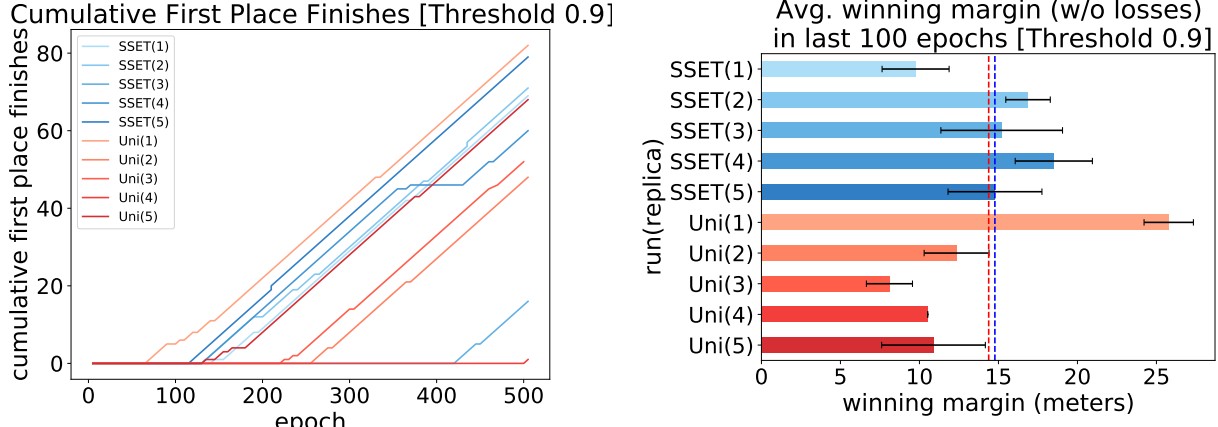

Figure 16: **Left:** Cumulative wins evaluated every 5 epochs in the slingshot passing test with a threshold of **0.9** (only counting slipstream from a very close opponent). In this case the difficulty of finding states with $> 0.9$ slipstream leads to higher variance, but not as high as uniform sampling. **Right:** Average (and std dev) of winning margins (excluding losses) in the last 100 epochs for each run.

from 70 epochs to 505 epochs to start winning. By contrast all 5 SSET runs learn to win consistently by epoch 110. In addition, the SSET runs seem to learn stronger passing skills, with an average winning distance (discarding runs where the agent lost) of 18.4 meters in the last 100 epochs compared to 14.4 meters for uniform sampling.

The behavior is somewhat sensitive to the choice of slipstream threshold. Figure 15 provides results in cases where the slipstream event was triggered by values greater than 0.1, which occurs very commonly when there is an opponent car ahead of the agent within a 60 meter range. Under these conditions, the event is not as informative as the $> 0.7$ event used earlier. Figure 15 shows that under these conditions, SSET still has better average performance and less variance than uniform sampling, but its performance is not as good as the results in Figure 14.

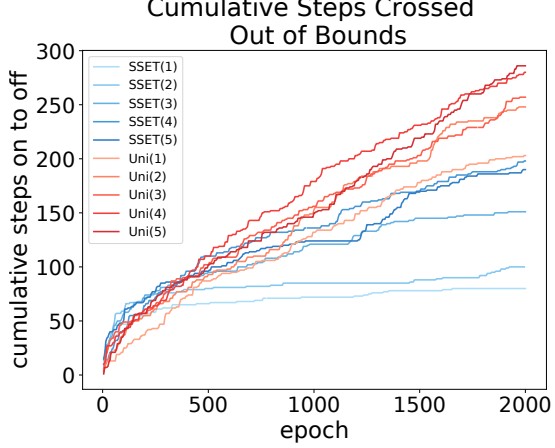

| Metric (epochs 1000-2000) | SSET | Uni |
|---|---|---|
| Min/Avg/Max fraction policies left track | .040 .111 .180 | .210 .253 .305 |
| Avg ± (σ) best lap time (s) w/o disqualified policies | 114.607 ± .12 | 114.459 ± .12 |
| Avg ± (σ) best lap time (s) with disqualified policies | 114.618 ± .15 | 114.496 ± .22 |

Figure 17: **Left:** Cumulative times an agent went off-course in 3-lap evaluations performed every 5 epochs. Uniform sampling runs (red lines) display catastrophic forgetting, oscillating between steady laps and policies that take risks and go off course. SSET converges to policies that more consistently stay on course (flatter blue lines). **Right**: Summary statistics on converged behavior. The worst SSET run stays on course more than the best uniform one, with comparable lap times.

Figure 16 illustrates results when a value greater than 0.9 was needed to trigger the event. This event requires significant exploration to trigger the event early in learning, since the agent must be very close to the car ahead to achieve such a value. Under these conditions, the variance of SSET increases significantly as the time it takes for the events to aid in learning is highly dependent on early exploration. However, SSET 's variance is still lower than the uniform sampling approach and SSET tends to win earlier and by a slightly larger margin.

Overall these results indicate that SSET is robust to events that happen frequently or (at the other extreme) are hard to find in early learning. SSET fares no worse than uniform sampling in these cases, though not as well as when using better chosen events.

## 7.2   Remembering to Stay On Course

We now present an experiment on maintaining multiple skills in a time-trial (solo car) setting on the full Lago Maggiore GP (Maggiore) track using a Porsche 911. The experimental details (see Section A.3) follow Wurman et al.'s except that we use only the time-trial reward components. The agent is tasked with running fast laps as well as avoiding off-course penalties. As learning progresses, off-course events become very scarce compared to the roughly 1200 steps an agent takes per lap. Consequently, learned behavior with a monolithic ERB and uniform sampling (red lines in Figure 17) oscillates: policies may not go off course for several epochs, then "forget" the potential penalty and shift to a policy that cuts corners, and the cycle repeats.

To retain consistent on-track behavior, we use SSET with a *re-establish* event that occurs if an agent returns to the track for 2 seconds after having left the course and use a history length of 7 seconds (roughly the half-life of the agent's horizon) to capture the full sub-trajectory of leaving and returning to the course. We set $\eta$ and $\kappa$ to 1% and used an ERB of total capacity 10-million. Note this is an extension of the formal event spec definition since the event condition here is based on a history of states, not just $s'$. The results of this approach (blue lines in Figure 17) show the oscillating behavior is replaced by consistent on-course laps, with 88.9% of SSET policies (up from 74.7% of uniform sampling policies) incurring no penalties after epoch 1000. Notably the *worst* off-course percentage for the SSET runs is still better than the *best* percentage with uniform sampling. The minimum (out of 3 in each eval) lap times averaged across later epochs are also within 0.15 seconds. A second set of 5 runs using a 1$s$ reestablish requirement with $\tau = 5s$ achieved virtually the same results (86.9% of policies on course with similar lap times). Thus, in a highly realistic

driving simulator with a deep neural network, SSET mitigates catastrophic forgetting and balances learning of multiple skills, echoing the results on MiniGrid domains in Sections 5.4 and 5.5.

## 8 Recommendations on How to Pick Helpful Events

SSET requires domain knowledge to specify the table partitions and in this regard, is similar to designing potential functions for shaping rewards. In large domains like Mujoco (Section 6) and GT (Section 7), there are already dense reward structures built into the canonical versions of the environments, so it is difficult to add "yet another reward term" in a meaningful way, but quite easy to use SSET and gain benefits from domain knowledge. As for the ease of specifying domain knowledge for SSET, we provide the following guidelines:

- For users that don't know their domain well, one can use a "goal" event (see mini-grid experiments in Section 5) or in a non-goal environment, use reward-threshold events (see Mujoco experiments in Section 6) with a fairly long history and reap benefits, even without understanding the true subgoals.

- Even if the events are poorly chosen, the technique usually does only as badly as uniform sampling (see Figures 9, 4, 5). The cases where it would actually hinder performance are relatively pathological (i.e.. using the majority of the buffer for incorrect events) and are easily avoidable in practice by setting reasonable caps on the event table sizes (say no more than 30% total as indicated in Figure 5 with table size experiments with bad events).

- Event Tables in SSET only require unsigned integer indices pointing to the data that is already stored in the main buffer. The additional memory footprint expands with the number of tables times their sizes. The publicly available Reverb (Cassirer et al., 2021) package already implements this data structure efficiently.

Beyond these guidelines for using user-specified events, the topic of learning a helpful set of events online is an open, but intriguing question. The difficulty in identifying events with no background knowledge is that by Definition 4 in the Appendix, events correspond to states that are most likely on optimal trajectories, so in the extreme case knowing the perfect set of events means knowing the optimal policy's state distribution. However, in cases where some model information is known, subgoal discovery methods could be used to propose events. One could also over-specify a large number of events and attempt to learn the weights ($\eta$ and $\kappa$) online, though the current work does not guarantee the convergence of such a meta-learner.

## 9 Conclusions and Future Work

This paper introduced Event Tables and SSET algorithm to improve the sample complexity and stability of off-policy RL algorithms. The theoretical results quantify the potential speedups and lend guidance for choosing event conditions. Experiments in MiniGrid, standard RL benchmarks, and Gran Turismo Sport show the benefits of SSET over monolithic ERBs or other prioritization schemes. Furthermore, combining TD-error PER or reward shaping on top of SSET led to further improvements in sample complexity and stability.

The current approach relies on domain experts or RL practitioners to designate event conditions. It may be possible to use subgoal discovery methods (Kulkarni et al., 2016; McGovern & Barto, 2001; Kompella et al., 2017) to automatically identify event conditions. Future studies may also consider dynamically changing the events or their proportions ($\eta$ and $\kappa$) based on learning progress. For instance, an event like "driving forward" might help early in learning but could be supplanted later by an "overtake" table as the agent hones in on more complex skills. Such dynamic rules, which might be based on the insertion rates in different tables, could be used to encode a curriculum that would directly change the complexion of mini-batches, not just the data collection tasks.

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

## A  Appendix

### A.1  Theoretical Analysis

In this section we prove sufficient conditions for the improvement of convergence speed using Event Tables and SSET. Specifically, we show that the more correlated events are with optimal behavior, the larger the expected improvement. To ground the analysis and make use of existing results, we use *tabular* Q-learning with a target function. We start from known finite-time convergence results in this setting (Lee & He, 2020; Li et al., 2022). Specifically, Theorem 1 from Li et al. (Li et al., 2022) (a refined version of Theorem 3 from Lee et al. (Lee & He, 2020)) computes the lower bound of sample complexity ($N$) for learning an $\varepsilon$-optimal solution using an iterative tabular Q-learning algorithm with target networks (Algorithm 2). We show that the lower bound of the sample complexity for achieving $\varepsilon$-optimal behavior is reduced by using event tables for sampling (Theorem 1).

The steps taken from the know result in the uniform sampling case to the new theorem are as follows. We define the state probability distribution (density) following a given policy to a finite horizon from an initial state and use that to define the state density disparity to the optimal policy (Definitions 2-3). We then formally define *event conditions* (Definition 4) on the states with low optimal-policy disparity or final states of the optimal policy. We extend this definition to event sections and their corresponding tables that include sufficient history to (on expectation) reach back to a previous event or initial state (Definitions 5-6). We then quantify the over-sampling of experience in the event tables (Lemma 1) and derive the convergence rate (Prop. 3) and bias correction procedure (Lemma 2). Finally, we show that the resulting convergence bound is an improvement over uniform sampling (Theorem 1).

---

**Algorithm 2:** Target Q-learning using an experience replay buffer $\mathcal{B}$

```
//input:  outer loop iteration number K, inner loop iteration number I, initialization Q_0,
step-sizes α_t.
//output:   Q_K
```

1 **for** $k = 0$ **to** $K - 1$ **do**
      // outer loop
2     $Q_{k,0} \leftarrow Q_k$
3     **for** $i = 0$ **to** $I - 1$ **do**
          // inner loop
4         $(s, a, r, s') \sim \mathcal{B}$  // randomly sample a transition
5         $Q_{k,i+1}(s,a) = Q_{k,i}(s,a) + \alpha_i(r + \gamma \max_{a'} Q_k(s',a') - Q_{k,i}(s,a))$  // update
6     **end**
7     $Q_{k+1} \leftarrow Q_{k,I}$  // refresh target
8 **end**

---

First, we provide a general overview of the base algorithm and the existing results.

**Assumptions**: Throughout the analysis we consider a finite discrete episodic MDP $M = (\mathcal{S}, \mathcal{A}, \mathcal{P}, \mathcal{R}, T, \gamma)$, where $\mathcal{S}$ and $\mathcal{A}$ are finite and discrete state and action spaces, $\mathcal{P}$ is the state transition probability and $\mathcal{R}$ is a bounded reward function, *s.t.* $r(s,a) \in [0,1]$, $\forall (s,a) \in \mathcal{S} \times \mathcal{A}$, $T$ is the length of each episode and $\gamma \in (0,1)$ is the discount factor. We assume for simplicity in notation that any terminal state traps the agent until time step $T$. We assume that the agent takes actions using a fixed stochastic behavior policy $\pi^b : \mathcal{S} \times \mathcal{A} \to [0,1]$, such that it can visit every state and execute every action with a non-zero probability within the $T$ time-step horizon. We denote $\pi^* : \mathcal{S} \times \mathcal{A} \to [0,1]$ to be an optimal policy of the MDP (Puterman, 2014), such that its state-action value function is optimal $Q^{\pi^*}(s,a) = \max_\pi Q^\pi(s,a)$, $\forall (s,a) \in \mathcal{S} \times \mathcal{A}$. Again for simplicity in notation, we assume that the event-tables are sampled uniformly with a total sampling probability denoted by $\eta = \sum_{i=1}^n \eta_i (= \eta/n)$ (see Alg. 1). The sampling probability of the default-table is $\eta_0 = 1 - \eta$.

Let $\mathcal{T} : \mathbb{R}^{|\mathcal{S}||\mathcal{A}|} \to \mathbb{R}^{|\mathcal{S}||\mathcal{A}|}$ define the Bellman operator

$$\mathcal{T}Q_k(s,a) = \mathbb{E}_{s' \sim P(.|s,a)} \left[ r(s,a) + \gamma \max_{a'} Q_k(s',a') \right].$$

Q-learning using a target-network $Q_k$ and an experience replay buffer $\mathcal{B}$ can be viewed as minimizing the following loss function

$$\min_{Q \in \mathbb{R}^{|\mathcal{S}||\mathcal{A}|}} l(Q; Q_k, \mathcal{B}) = \frac{1}{2} \mathbb{E}_{(s,a,r,\cdot) \sim \mathcal{B}} \left[ \left( \mathbb{E}_{s' \sim P(.|s,a)} \left[ r(s,a) + \gamma \max_{a'} Q_k(s',a') \right] - Q(s,a) \right)^2 \right] \quad (1)$$

For a given buffer $\mathcal{B}$ generated using the fixed behavior policy $\pi^b$, we define constants (following the notation used in (Lee & He, 2020; Li et al., 2022))

$$c_{\mathcal{B}} := \min_{s \in \mathcal{S}, a \in \mathcal{A}} P((s,a,\cdot,\cdot) \sim \mathcal{B})$$

$$L_{\mathcal{B}} := \max_{s \in \mathcal{S}, a \in \mathcal{A}} P((s,a,\cdot,\cdot) \sim \mathcal{B})$$

Here, $c_{\mathcal{B}}$ and $L_{\mathcal{B}}$ denote the minimum and the maximum probability that a state-action pair $(s,a)$ is sampled from the buffer. From our assumptions, $c_{\mathcal{B}} > 0$.

Carrying out $N$ steps of SGD optimizing Eq. 1, we get an approximation of $\mathcal{T}Q_k$ with a certain error bound $\mathbb{E}[\|Q_{k+1} - \mathcal{T}Q_k\|] \le \varepsilon_{k+1}$, which then accumulates across outer iterations over $k$.

**Proposition 1.** *(Theorem 1 (Li et al., 2022)) Consider an MDP with $Q_0 = 0$, $c_{\mathcal{B}} > 0$, $\alpha_t = \frac{\alpha}{\lambda + t}$, where $\alpha = 2/c_{\mathcal{B}}$ and $\lambda = (13\gamma^2 L_{\mathcal{B}})/(2c_{\mathcal{B}}^2)$. The minimum number of samples required to achieve an $\varepsilon$-optimal solution $\mathbb{E}[\|Q_K - Q^*\|_\infty] \le \varepsilon$ using Algorithm 2 is given by*

$$N^{\mathcal{B},K} = \frac{832\gamma^2}{(1-\gamma)^5 \varepsilon^2} \log \left( \frac{4}{(1-\gamma)\varepsilon} \right) \frac{L_{\mathcal{B}}}{c_{\mathcal{B}}^3}.$$

We show SSET can decrease this bound when the events are correlated with an optimal policy and histories are sufficiently long. To quantify these conditions, we use the following definitions of optimal trajectories and their state densities induced by various policies.

**Definition 1.** *Trajectory: We define a trajectory $\Gamma_{s_i, s_j}^\pi$ of the MDP M as a temporal sequence of transition tuples*

$$\Gamma_{s_i, s_j}^\pi = \left\{ (s_k, a_k, r_k, s_{k+1}) \mid s_{k+1} \sim P(\cdot|s_k, a_k \sim \pi(\cdot|s_k)), r_k \sim \mathcal{R}_{a_k}^{s_k, s_{k+1}}, \forall k \in [i, j-1]) \right\}.$$

*For simplicity, $|\Gamma_{s_i, s_j}^\pi|$ denotes the length or the number of transitions of the trajectory.*

**Definition 2.** *State Density: We define $\rho^{\pi, s_0, K} : \mathcal{S} \to [0, 1]$ as the state probability distribution following any policy $\pi$ for the MDP with the initial state $s_0$ over a finite time horizon $K$*

$$\rho^{\pi, s_0, K}(s) = \frac{1}{C} \sum_{k=0}^K P(s_k = s | \pi, s_0). \quad (2)$$

*$C$ enforces the constraint $1^T \overline{\rho}^{\pi, s_0, K} = 1$ to make $\rho^{\pi, s_0, K}$ a probability distribution, where $\overline{\rho}^{\pi, s_0, K} = [\rho^{\pi, s_0, K}(s_1), ..., \rho^{\pi, s_0, K}(s_N)]^T \in \mathbb{R}^N$.*

**Definition 3.** *State Density Disparity from Optimal: We define $\widetilde{\rho}^{\pi, s_0, K} : \mathcal{S} \to [-1, 1]$ as the difference in the state density following any policy $\pi$ compared to an optimal-policy $\pi^*$ for the MDP with the initial state $s_0$ over a finite time horizon $K$*

$$\widetilde{\rho}^{\pi, s_0, K}(s) = \rho^{\pi^*, s_0, K}(s) - \rho^{\pi, s_0, K}(s). \quad (3)$$

We now formally define an *event condition* $\omega : \mathcal{S} \to \{0, 1\}$, which is the core concept of our SSET algorithm. An event occurs when an agent enters a state that satisfies the event condition, which implicitly defines a set of event states: $\mathbb{I}_{[s \in \mathcal{S}^{\omega_i}]}$. Intuitively, good event conditions should be aligned with the optimal policy and also act as waypoints linking initial and final states visited by the optimal policy. Therefore, final states visited by the optimal policy should also satisfy at least one event condition.

**Definition 4. Event condition**: *Let $\mathcal{I}$ denote the initial state distribution of the episodic MDP with the episode length $T$. For a given threshold $\mu \in (0, 1)$, we define a collection of event sets $\mathcal{S}^\omega = \{\mathcal{S}^{\omega_1}, ..., \mathcal{S}^{\omega_n}\}$, s.t. $\forall s_i \in \mathcal{S}^{\omega_i} \; \exists s_j \in \mathcal{I} \cup \mathcal{S}^{\omega_j}$ where the following conditions are true:*

$$either \qquad \overset{\sim}{\rho}^{\pi^b, s_j, T}(s_i) \geq \Delta = (1 - \mu) \qquad (4)$$

$$or \quad |\Gamma^{\pi^*}_{s \in \mathcal{I}, s_i}| = T \qquad (5)$$

*We define an event condition $\omega_i : S \to \{0, 1\}, \forall i \in [1, n]$ s.t. $\omega_i(s) = \mathbb{I}_{[s \in \mathcal{S}^{\omega_i}]}$.*

Intuitively Condition (4) covers states that are significantly (based on $\mu$) more likely under the optimal policy than under $\pi^b$. Condition (5) covers terminal states visited by the optimal policy in case they are not satisfied under Condition (4). Note, the definition above could cover a large amount of the state space in highly stochastic domains. Therefore, Theorem 1 filters the set further to focus on higher probability states without significant degradation to the overall sample complexity.

For simplicity in the notation used in our proofs, we denote an event section $\mathcal{E}^{\omega_i}$ as a set of states whose state density disparity from the optimal policy is less than (or equal to) $\Delta$. Intuitively, these are the states from which the agent can easily reach the event states.

**Definition 5. Event Section**: *We define an event section $\mathcal{E}^{\omega_i}$ as*

$$\mathcal{E}^{\omega_i} = \left\{ s \; \middle| \; \sup_{s' \in \mathcal{S}} \left\{ \overset{\sim}{\rho}^{\pi^b, s', T}(s_i), \; \forall s_i \in \mathcal{S}^{\omega_i} \right\} = \Delta \right\}.$$

**Proposition 2.** *For a given $\mu$, $\exists \mathcal{S}^\omega$ s.t. all initial states of the MDP $\mathcal{I}$, the event states $\mathcal{S}^{\omega_{\forall i}}$, and the optimal terminal states $\{s \mid |\Gamma^{\pi^*}_{s_0 \in \mathcal{I}, s}| = T\}$) belong to at least one event section $\mathcal{E}^\omega$.*

*Proof.* Let us first consider the case where for a given $\mu$, Condition (4) does not hold for any $s \in \mathcal{S}$. From Def. 4, $\mathcal{S}^\omega$ contains only a single event-set that includes all the optimal terminal states $\mathcal{S}^{\omega_{\text{term}}} = \{s \mid |\Gamma^{\pi^*}_{s_0 \in \mathcal{I}, s}| = T\}$). Therefore, from Def. 5, all initial states belong to the event section $\mathcal{E}^{\omega_{\text{term}}}$, and hence the result is true for this case.

Now for the case where there are non-zero number of event-sets that satisfy Condition (4), it follows that the event states belong to either the event section where Condition (4) is true or the terminal $\mathcal{E}^{\omega_{\text{term}}}$. $\square$

From the definition of events, we can define an event table that stores these experiences that lead to them.

**Definition 6. Event Table**: *An event table $\mathcal{B}^{\nu_i}$ for event spec $\nu_i = \langle \omega_i, \tau_i \rangle$, denotes an experience replay buffer, which is a multiset (a set with repeated elements) of transitions from trajectories of a given maximum length $\tau_i$ s.t.*

$$\mathcal{B}^{\nu_i} = \bigcup \left\{ (s, a, r, s') \mid (s, a, r, s') \in \Gamma^{\pi^b}_{s_i, s^{\omega_i}}, |\Gamma^{\pi^b}_{s_i, s^{\omega_i}}| \leq \tau_i, \forall s^{\omega_i} \in \mathcal{S}^{\omega_i} \right\}. \qquad (6)$$

Using the above definitions, we can now analyze the likelihood of sampling any particular state from an ERB that contains both event tables and a default table $\mathcal{B}^0$ based on fixed sampling probabilities of the event tables (as used in Algorithm 1). The following lemma quantifies the over-sampling of experiences in the event tables.

**Lemma 1.** *Let $\mathcal{B}^\nu = \underset{\forall i \in [1, n]}{\cup} \mathcal{B}^{\nu_i}$ denote the union events table and $\mathcal{B}^0$ denote the default table that contains all the transitions collected following a fixed behavior policy $\pi^b$. $s \overset{\eta}{\sim} \mathcal{B}^\nu \cup \mathcal{B}^0$ denotes a weighted sampling*

$(0 < \eta < 1)$ of a transition tuple $(s, \cdot, \cdot, \cdot)$ between the event and the default tables. For any event-spec $\nu_i$,

$$\text{if} \quad \tau_i \leq \frac{(1-\eta)^m}{(m+1)n\mu},$$

$$\text{then} \quad P(s \overset{\eta}{\sim} \mathcal{B}^\nu \cup \mathcal{B}^0) \geq (1-\eta)^{-m} P(s \sim \mathcal{B}^0), \ \forall (s, \cdot, \cdot, \cdot) \in \mathcal{B}^{\nu_i}, m \in \mathbb{Z}^{0+}.$$

*Proof.* Expanding the weighted sampling probability, we have

$$P(s \overset{\eta}{\sim} \mathcal{B}^\nu \cup \mathcal{B}^0) = \eta P(s \sim \mathcal{B}^\nu) + (1-\eta)P(s \sim \mathcal{B}^0)$$

$$= \eta P(s \sim \mathcal{B}^\nu) + (1-\eta)P(s \sim \mathcal{B}^0) - \frac{P(s \sim \mathcal{B}^0)}{(1-\eta)^m} + \frac{P(s \sim \mathcal{B}^0)}{(1-\eta)^m}$$

$$= \eta \left( P(s \sim \mathcal{B}^\nu) - \frac{1 - (1-\eta)^{m+1}}{\eta(1-\eta)^m} P(s \sim \mathcal{B}^0) \right) + (1-\eta)^{-m} P(s \sim \mathcal{B}^0)$$

$$= \eta \left( P(s \sim \mathcal{B}^\nu) - \frac{\sum_{k=0}^m (1-\eta)^k}{(1-\eta)^m} P(s \sim \mathcal{B}^0) \right) + (1-\eta)^{-m} P(s \sim \mathcal{B}^0)$$

$$\geq \eta \left( P(s \sim \mathcal{B}^\nu) - \frac{m+1}{(1-\eta)^m} P(s \sim \mathcal{B}^0) \right) + (1-\eta)^{-m} P(s \sim \mathcal{B}^0) \quad (\because (1-\eta) < 1)$$

$$\overset{\because (s,\cdot,\cdot,\cdot) \in \mathcal{B}^{\nu_i}}{\underset{Def.\ 6}{=}} \eta \sum_{k=0}^{\tau_i} P(s_{\tau_i - k} = s | s_{\tau_i} = s^{\omega_i}, \pi^b) \left[ P(s_{\tau_i} = s^{\omega_i} | \mathcal{B}^\nu) - \frac{m+1}{(1-\eta)^m} P(s_{\tau_i} = s^{\omega_i} | \mathcal{B}^0) \right]$$

$$+ (1-\eta)^{-m} P(s \sim \mathcal{B}^0), \quad s^{\omega_i} \in \mathcal{S}^{\omega_i}. \tag{7}$$

$\mathcal{B}^{\nu_i}$ contains only trajectories of length $\leq \tau_i$ that all lead to $s^{\omega_i} \in \mathcal{S}^{\omega_i}$. Therefore $P(s_{\tau_i} = s^{\omega_i} | \mathcal{B}^{\nu_i}) \geq \frac{1}{\tau_i}$. Given the event-table sampling algorithm described in Sec. 4 (with the assumption that a table is uniformly sampled), we have $P(s_{\tau_i} = s^{\omega_i} | \mathcal{B}^\nu) = P(\omega_i \sim \{\omega_i, \forall i\}) P(s_{\tau_i} = s^{\omega_i} | \mathcal{B}^{\nu_i}) \geq \frac{1}{n\tau_i}$. For the event section $\mathcal{E}^{\omega_i}$ from Def. 4 and $\rho^{\pi^*, \cdot, T}(s^{\omega_i}) \leq 1$, therefore $P(s_{\tau_i} = s^{\omega_i} | \mathcal{B}^\nu) \leq \mu$. Substituting in Eq. 7, we get

$$P(s \overset{\eta}{\sim} \mathcal{B}^\nu \cup \mathcal{B}^0) \geq \eta \sum_{k=0}^{\tau_i} P(s_{\tau_i - k} = s | s_{\tau_i} = s^{\omega_i}, \pi^b) \left[ \frac{1}{n\tau_i} - \frac{m+1}{(1-\eta)^m} \mu \right] + (1-\eta)^{-m} P(s \sim \mathcal{B}^0)$$

$$\geq (1-\eta)^{-m} P(s \sim \mathcal{B}^0) \quad \because \frac{1}{n\tau_i} \geq \frac{m+1}{(1-\eta)^m} \mu$$

$\square$

Typically, the threshold $\mu$ is very small as the state density following the behavior policy drops exponentially with the number of forward time steps. With small values of $\mu$, next we show that $m > 1$, thereby quantifying the over-sampling of experiences in positive powers of $\frac{1}{(1-\eta)}$.

**Proposition 3.** *$m$ is asymptotically convergent to $-\ln(\tau_i \mu) - \ln\ln(\tau_i \mu) + o(1)$ as $\mu \to 0$.*

*Proof.* Rearranging the condition in Lemma 1 for equality

$$e^{(m+1)\ln(1-\eta)} = \frac{(1-\eta)n\tau_i\mu}{\ln(1-\eta)}(m+1)\ln(1-\eta)$$

Setting $a = 1/(1-\eta)$, $b = |nu|\tau_i\mu/a$, $x = (m+1)$ and $y = (x \ln a)$, the equation reduces to $ye^y = \frac{\ln a}{b}$. The solution to this equation is the Lambert W function $W_0(\frac{\ln a}{b})$, since $\frac{\ln a}{b} > 0$

$$m = \frac{1}{\ln(a)} W_0 \left( \frac{a \ln(a)}{n\tau_i\mu} \right) - 1. \tag{8}$$

$W_0(x)$ is asymptotic to $\ln x - \ln \ln x + o(1)$ for large values of $x$ (Corless et al., 1996). With small values of $\mu$, we get

$$m = -\ln(\tau_i \mu) - \ln \ln(\tau_i \mu) + o(1). \tag{9}$$

$\square$

Sampling using event-tables may introduce a bias as it can change the expected value of the stochastic Bellman operator $\mathcal{T}Q_k(s,a) = \mathbb{E}_{s' \sim P(.|s,a)} \left[ r(s,a) + \gamma \max_{a'} Q_k(s',a') \right]$ for the $(s,a)$ pair whose next state has a finite probability to either belong to the event-tables or not. We can correct this bias by computing weights for weighted importance sampling (Mahmood et al., 2014) similarly to PER (Schaul et al., 2016).

**Lemma 2.** *Bias introduced by the weighted sampling $s \overset{\eta}{\sim} \mathcal{B}^\nu \cup \mathcal{B}^0$ between the event and the default tables is given by*

$$w(s,a) = \begin{cases} 1 - \eta \sum_{s' \in \mathcal{S}} P((s,a,\cdot,s') \notin \mathcal{B}^\nu | \, s,a), \\ \qquad\qquad\qquad\qquad\qquad\quad if \, (s,a,\cdot,\cdot) \in \mathcal{B}^\nu \\ \frac{1}{1-\eta}, \qquad otherwise. \end{cases}$$

*Proof.* Event tables construction (see Algorithm 1) prioritizes transitions that are in either of the event buffers over the ones that are out. Given an $(s,a)$ pair, the transitions that are not present in any of the event tables are sampled from the default table with a probability of $1 - \eta$. Therefore, the remainder $\eta \sum_{s' \in \mathcal{S}} P(s' \notin \mathcal{B}^\nu | \, s,a)$ is adjusted among the probabilities of transitions in the event-buffers. The correction weight that is needed is $1 - \eta \sum_{s' \in \mathcal{S}} P(s' \notin \mathcal{B}^\nu | \, s,a)$. For the transitions that are not in the event buffers, only a scaled correction of $(1 - \eta)$ is required. $\square$

Now that we have quantified the oversampling and bias-corrections of event histories, we state our main theorem showing that with sufficient history and events that are correlated with optimal behavior, the convergence speed of Q-learning to an optimal policy is improved over the monolithic ERB ($\mathcal{B} = \mathcal{B}^0$) sample complexity ($N^{\mathcal{B},K}$).

**Theorem 1.** *Let $\mathcal{S}^f$ denote the set of states s.t. the sampled optimal trajectories starting from those states, are contained in the combined event-buffer with a probability greater than $\bar{p} \in (0,1]$*

$$\mathcal{S}^f = \{s \mid P(\Gamma^{\pi^*}_{s,s'} \subset \mathcal{B}^\nu) \geq \bar{p}\}.$$

*Under the conditions of Prop. 1 and if $\tau_{\forall i \in [1,n]} \leq \dfrac{(1-\eta)^m}{(m+1)n\mu}$, then*

$$P\left( N^{\mathcal{B}^\nu \cup \mathcal{B},K} \leq (1-\eta)^{2m} N^{\mathcal{B},K} \right) \geq \bar{p}, \; \forall s \in \mathcal{S}^f, m \in \mathbb{Z}^{0+}.$$

*Proof.* Let $M^{\mathcal{S}^f} = (\mathcal{S}^f, \mathcal{A}, \mathcal{P}, \mathcal{R})$ be a reduced MDP of $M$ with the initial, event and terminal states in $\mathcal{S}^f$ (Prop. 2). With the bias correction applied from Lemma 2, $\pi^{*,M^{\mathcal{S}^f}}(s) = \pi^{*,M}(s), \forall s \in \mathcal{S}^f$. With $c^{\mathcal{B}^\nu \cup \mathcal{B}^0} := \min_{s \in \mathcal{S}^f, a \in \mathcal{A}} P((s,a,\cdot,\cdot) \sim \mathcal{B}^\nu \cup \mathcal{B}^0)$ and $L^{\mathcal{B}^\nu \cup \mathcal{B}^0} := \max_{s \in \mathcal{S}^f, a \in \mathcal{A}} P((s,a,\cdot,\cdot) \sim \mathcal{B}^\nu \cup \mathcal{B}^0)$ as constants the result of the theorem then follows from using Lemma 1 in Prop. 1 for MDP $M^{\mathcal{S}^f}$ using event-tables for experience replay. $\square$

**Remark 1.** *Applying the results of Theorem 1 and Lemma 1 in the loss function of Q-learning (Eq. 1), the speed-up $(1-\eta)^{-2m}$ compounds across multiple outer iterations of target Q-learning (Alg. 2). Therefore, the transitions that are not in the event buffers are also benefited in expectation, countering the lower sampling rate of $(1-\eta)$.*

### A.2 Learning Parameters and Resources Used in MiniGrid and Continuous Control Experiments

Table 1 lists the parameters used in our MiniGrid experiments. Event conditions are true when the agent interacts with an object, the goal, or crosses between rooms. History lengths are shorter in the obstacle course where the events are denser (since there are more objects), but we use a larger buffer there to accommodate the more diverse skills and larger number of event tables. For all the MiniGrid experiments, we used two asynchronous rollout workers each with 1.5 CPUs and 2GB memory to collect and transmit experience data to a replay buffer efficiently implemented using Reverb (Cassirer et al., 2021). The training was conducted using two virtual CPUs and 3 GB of memory at a rate of 40 training steps per sec. For the continuous control tasks (LunarLanderContinuous and MuJoCo domains), we used a single rollout worker with 1 CPU and 2GB memory, and the training was conducted asynchronously using 7.7 CPUs with 8GB of memory. Table 2 lists all the algorithm parameters used for these benchmark experiments. Parameters that are not related to SSET are either set to values that are typically used in the literature (*e.g.* (Hong et al., 2021)) (discount factor, stale network refresh rate, max buffer capacity, batch size) or set based on parameter tuning to get the best performance of the baselines (learning-rates, TD-priority exponent, architectures, epsilon-greedy).

Table 1: Parameters in the MiniGrid domain

| Parameter | Three Room Grid World | Obstacle Course |
|---|---|---|
| Event conditions ($\omega_i$) | at-gap, done | at-spike, at-lava, at-gap, pickup-key, at-door, done |
| Event history length ($\tau$) | 200 | 50 |
| Event sampling probabilities ($\eta_i$) | Default: 0.5, at-gap: 0.2, done: 0.3 | Default: 0.5, at-spike: 0.1, at-lava: 0.0625, at-gap: 0.0625, pickup-key: 0.0625, at-door: 0.1125, done: 0.1 |
| Max buffer capacity | 20000 | 100000 |
| Value function networks | 1 hidden layer of 256 ReLU units | 2 hidden layers of 256 ReLU units each |
| Learning rate | 1e-3 | 5e-4 |
| Table capacity sizes ($\kappa_i$) | max buffer capacity * $\eta_i$ | |
| batch-size | 32 | |
| epsilon-greedy ($\epsilon$) | 0.3 | |
| TD-Priority exponent | 0.65 | |
| Stale network refresh rate | 0.01 | |
| Discount factor | 0.99 | |

### A.3 Appendix: Details of Gran Turismo Experiments

The Gran Turismo™ (GT) Sport (`https://www.gran-turismo.com/us/`) racing simulator for PlayStation™ 4 allows an agent to race automobiles with highly realistic dynamics. The environment was recently used in (Wurman et al., 2022) to demonstrate an RL system that learned to beat human e-Sports champions on 3 different tracks in 4v4 (4 humans and 4 RL agent) competitions. Our experiments investigate smaller scenarios, either on sections of a track or without opponents, in order to isolate the specific effects of Event Tables . Almost all of the parameters of the representation and learning algorithms are the same as those reported in (Wurman et al., 2022) for the selected track / car combinations. We provide a summary of these settings below and note the small deviations for our particular scenarios.

The first experiment investigated is a slingshot passing scenario on a 1700m. straightaway at the Circuit de la Sarthe (Sarthe) track, using a Red Bull X2019 Competition race car, similar to a Formula 1 vehicle (see Figure 18). Training is performed in a one-on-one race against a built-in AI opponent from the game with

Table 2: Parameters in the Continuous Control Benchmark Tasks

| Parameter | LunarLanderContinuous | MuJoCo |
|---|---|---|
| Event conditions ($\omega_i$) | land-between-flags (lf), land-near-middle (lm) | Salient reward thresholds
Half-Cheetah: ($r > 8$, $r > 12$, $r > 16$)
Hopper: ($r > 2.5$, $r > 3$, $r > 3.5$)
Walker2D: ($r > 4$, $r > 5$, $r > 6$, $r > 7$)
Humanoid: ($r > 5$, $r > 7$, $r > 10$) |
| Event sampling probabilities ($\eta_i$) | Default: 0.7, lf: 0.1, lm: 0.2 | Default: 0.6,
Half-Cheetah: ($r_8$: 0.2, $r_{12}$: 0.1, $r_{16}$: 0.1)
Hopper: ($r_{2.5}$: 0.2, $r_3$: 0.1, $r_{3.5}$: 0.1)
Walker2D: ($r_4$: 0.2, $r_5$: 0.1, $r_6$: 0.05, $r_7$: 0.05)
Humanoid: ($r_5$: 0.2, $r_7$: 0.1, $r_{10}$: 0.1) |
| Max buffer capacity | 20000 | 1000000 |
| batch-size | 32 | 256 |
| Learning rates | Actor: 0.0003, Critic: 0.0003 | |
| SAC networks | 2 hidden layer of 256 ReLU units each, Gaussian Policy | |
| Target entropy (optimized entropy) | -2.0 | Half-Cheetah: -6.0
Hopper: -3.0
Walker2D: -6.0
Humanoid: -17.0 |
| Table capacity sizes ($\kappa_i$) | max buffer capacity * $\eta_i$ | |
| Event history length ($\tau$) | 200 | |
| Stale network refresh rate | 0.005 | |
| Discount factor | 0.99 | |

the RL agent always starting behind. To succeed, the RL agent needs to use the opponent's slipstream to accelerate beyond its top speed in open air and use the added momentum to "slingshot" past the opponent and hold it off to the end of the section. The second setting is a time trial competition on the full Lago Maggiore GP (Maggiore) track using a Porsche 911 from the FIA GT3 class of cars (see Figure 19). There the experiment focuses on driving fast lap times without going off course.

We used the same state features as the prior work in GT. State features include information about the agent's velocity, acceleration, tire slip, most recent actions (steering, throttle, and brake), position on the track, and the "course points" outlining the shape the upcoming track. Indicator features capture collisions with the walls or other cars as well as going off course by more than two tires (the game's definition of leaving the track). A $[0, 1]$ *slipstream* feature provided by the game measures the draft from car(s) ahead is also passed in as a state feature. Opponent cars were represented by two lists, one for opponents ahead, and one for opponents behind, with opponents masked out if they are more than 20 meters behind or 75 m. ahead. Each opponent in view was represented with their agent-centered relative position, velocity, and acceleration. The agent controlled the car by sending actions for the the steering wheel and a combined throttle/brake dimension.

The previous work with Gran Turismo used different reward functions on different tracks and different scenarios (such as Time Trial racing and head-to-head competition). We used the reward functions from their investigation that best fit our scenarios. In our slingshot passing tests at the Sarthe track, we used the reward components previously used in 4v4 competitive racing. These components and their weights are specified in Table 3 and included a reward for forward progress, penalties for hitting the wall, and a penalty for going off course that was linearly proportional to the agent's velocity. A passing reward provided reward for approaching an opponent from behind or pulling away from ahead (and penalized the opposites). For car collisions, three different components were used to penalize different aspects of collisions, including hitting

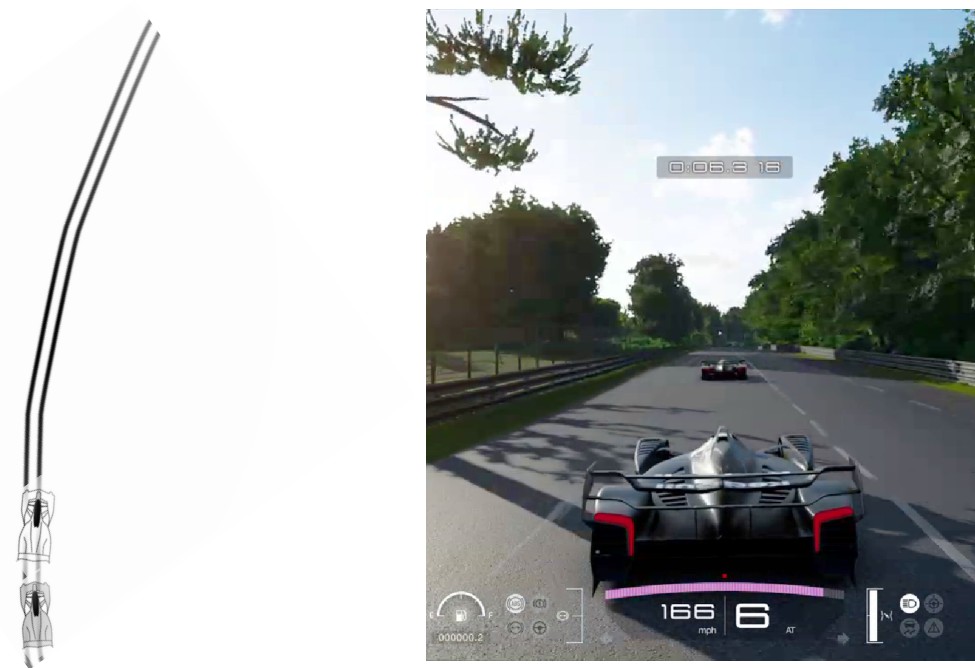

Figure 18: The section of the Sarthe track used for slingshot passing and a screenshot from the experiment.

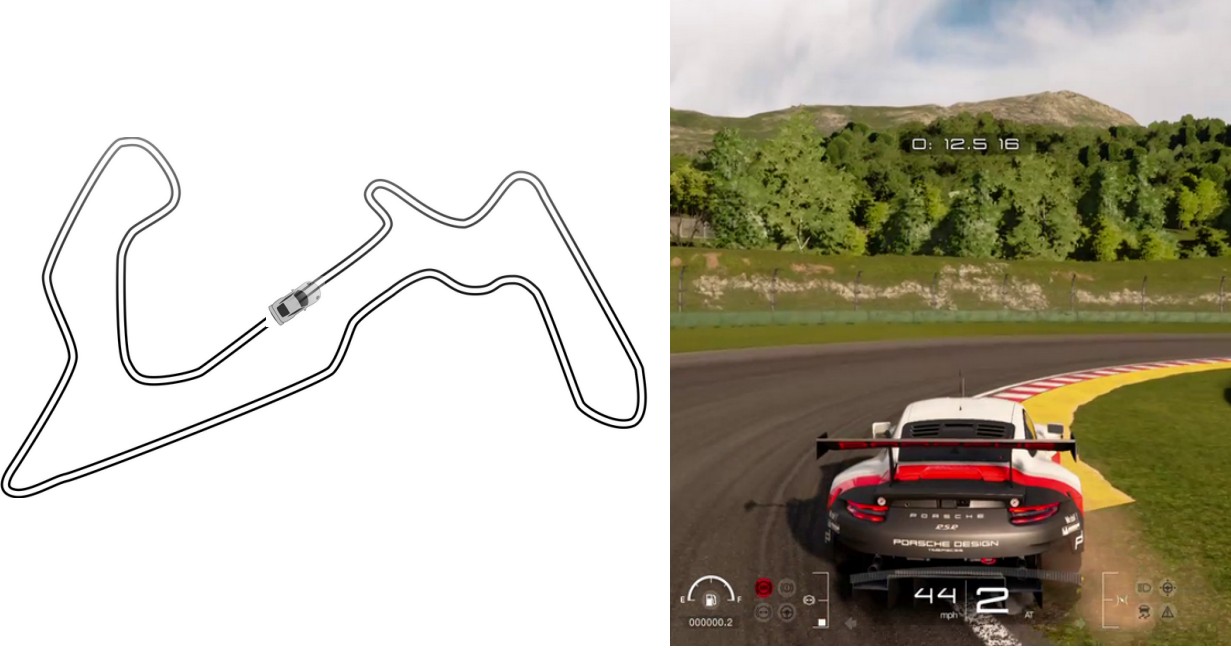

Figure 19: The Maggiore track used for Time Trial training and a screenshot from the experiment.

cars from behind, at-fault collisions, and a penalty for any collision at all. Because the Slingshot experiment focused on a section without difficult curves we did not use the "mistake learning" spin-out replays that prior work used on the Sarthe track.

For the off-course experiments at Maggiore, we used the reward components from previous work at that track but dropped the reward terms pertaining to other cars (such as collision or passing). These components

Table 3: Reward components used in the two experiments in the paper, slingshot passing and time trial training. The reward weights are all the same as (Wurman et al., 2022)'s results but the car collision and passing components are dropped in the time trial scenario.

| Reward Component | weight in slingshot (Sarthe) | weight in off-course TT (Maggiore) |
|---|---|---|
| Progress | 1 | 1 |
| Off-course$^2$ | 0 | 0.01 |
| Off-course-linear | 5 | 0 |
| wall penalty | 0.1 | 0.1 |
| tire slip (per tire) | 0 | 0.25 |
| passing bonus | 0.5 | 0 |
| car collision (any) | 4 | 0 |
| car collision (rear) | 0.1 | 0 |
| car collision (unsportsmanlike) | 5 | 0 |

include a wheel-slip penalty and an off-course penalty based on the square of the agent's velocity, which replaces the linear off-course penalty from Sarthe.

The base RL algorithm and all of its parameters were kept the same as Wurman et al.'s experiments. The base RL algorithm was Quantile Regression Soft Actor Critic (QR-SAC), a version of SAC where the critic networks represent the quantiles of the value function rather than just their mean. The neural networks used for value function and policy networks were all feed-forward networks with 2048 ReLU units in each of 4 hidden layers. Mini-batches of size 1024 were used with 6000 mini-batches per epoch. These samples were pulled from an ERB with total capacity ($\sum \kappa_i$) of 2.5 million (slingshot test on a 1.7 kilometer segment) or 10 million (time trial test on a nearly 6 kilometer track). Each table was blocked from sampling until it had at least 1024 experiences in the time trial test or 5000 experience tuples in the slingshot test (avoiding over-fitting). The Adam optimizer was used to optimize the weights with learning rates of $5 \times 10^{-5}$ for the $Q$-networks and $2.5 \times 10^{-5}$ for the policy network. A discount factor of 0.9896 was used and the SAC entropy temperature controlling exploration was 0.01. Dropout (0.1) was used when learning the policy network.

Following the hardware setup from the prior work, both of our experiments used 21 PlayStations in parallel during training with 1 of those PlayStations typically devoted to evaluations. In the slingshot experiments on a small segment of the track, each PlayStation had only one agent racing against an opponent. The competitor in this scenario was the game's own built-in AI with randomization controlling the spacing of the agents (uniformly drawn in $[10, 40]$ meters) and lateral spacing. The learning agent always starting behind the opponent. To provide extra diversity, the *Balance of Power* on the opponent, that is its horsepower and weight, were randomly increased or decreased up to 25% in each training episode. Training episodes were capped at 60 seconds although less than 25 seconds were usually sufficient to complete the section (also ending the training episode).

For the time-trial training scenario at Maggiore, no opponents were needed but we utilized the parallel collection strategy from (Fuchs et al., 2021; Wurman et al., 2022) to spawn 20 different agents uniformly around the track and collect data from each of these, yielding roughly 400 experiences per time step. Training episodes in these scenarios lasted 150 seconds. Notice that in both experiments there is only one training scenario with some minor randomization on exact locations, so we did not need to employ a task sampling scheme like the one needed to master the full racing scenario.

Using the same setup as prior work, agents on the PlayStations were controlled by rollout workers using two virtual CPUs and 3.3 GB of memory at a frequency of 10Hz. Policies were sent from the trainer to the rollout workers at the beginning of each training episode and kept static until the next episode. Actions and observations were sent between the rollout worker and the PlayStation through a restricted API . Experience was periodically streamed back to a trainer and stored in an ERB implemented via Reverb (Cassirer et al., 2021). Training was conducted using one NVIDIA V100 or half of an NVIDIA A100, ~8 virtual CPUs and 55 GB of RAM.

