# OpenReview forum: "Event Tables for Efficient Experience Replay"
_TMLR — Accepted by TMLR_

### Review · Reviewer_CmcH · 2023-01-19

**Summary Of Contributions:**

This paper investigates the best way of sampling experience replay for learning reinforcement learning agents in contrast to the standard methods of uniform or prioritized sampling from the replay buffer. They propose to define "events" that characterize behaviors in MDPs and use those to help guide how the replay sampling should be done (alg 1) and show extensive results improving upon the standard environments, including MiniGrid, Mujoco, Lunar Lander, and a Gran Turismo Sport racing simulator.

**Audience:**

Yes

**Claims And Evidence:**

Yes

**Requested Changes:**

I only have one question: are there other ways of integrating events than just in how the replay buffer is sampled? it seems like they could be used to create a curriculum or other kind of reward augmentation

**Strengths And Weaknesses:**

Strengths
+ This is a great paper! I like the idea of using events for sampling from the replay and this paper convincingly shows how using them for replay sampling significantly improves the training of RL agents in many non-trivial MDPs in contrast to uniform and prioritized sampling. The claims of this helping are strongly backed up.
+ The paper is well-written and clearly communicates all of the key ideas, insights, and recommendations. I especially like section 8 with recommendations on picking helpful events
+ The concept of events in MDPs seems like a very reasonable way of integrating more domain knowledge into MDP solutions.

Weaknesses
+ While events are nice way of integrating more knowledge into MDPs, they require manually specifying more information about how the agent should explore and learn an optimal policy. This gives the event-driven settings a slight advantage in comparison to the baselines that are not able to use this information.

---

> ### Author Response · Authors · 2023-03-18
> **Response to Reviewer CmcH (R1)**
>
> Thank you very much for your comments on the paper.  In answer to your question about whether events could be used in a curriculum context, we believe the answer is “yes”.  A simple version of such a curriculum is realizable in the current algorithm based on the $d_i$ parameters.  These parameters state that an event table should not be used in mini-batch construction until it has at least $d_i$ elements.  Usually that parameter is set to a small number just to make sure that, say, a partition with 5 entries isn’t oversampled.  But one could imagine setting $d_i$ much higher, which would essentially encode a curriculum that will begin adding certain experiences to training based on the agent’s progress.  One could then go further and instead base the condition not on $d_i$ but instead on having reached an event that often precedes this table or even turning “off” a table after other tables have become active.  Such control is an advanced topic but shows that Event Tables could be used to encode a curriculum. We added some extra notes to the conclusion addressing this possible extension.

---

### Review · Reviewer_j8j6 · 2023-01-20

**Summary Of Contributions:**

This paper presents Stratified Sampling from Event Tables (SSET) that introduces multiple "event tables" which are separate replay buffers that contains trajectory segments containing a human-defined "event" states and does stratified sampling from them. A major difference to previous works is that (i) SSET stores histories of states that lead to specific events instead of storing each state and (ii) SSET can utilize any state other than initial state for partitioning the buffer.  Experiments evaluate SSET on various domains, including GridWorld, Gym benchmark tasks, and a simulated racing game.

**Audience:**

Yes

**Claims And Evidence:**

No

**Requested Changes:**

My suggested changes are mostly focused on experiments:
- Investigation into the importance of history length ($\tau$) is required on multiple domains as it's currently reported when the event is not intermediate (Figure 7). From the current experimental results, it's not clear whether it's important or not.
- Invesitgation into the sensitivity to event sampling probabilities ($\eta_{i}$) is required as it's a quite complex additional hyperparameter for SSET. Is the method robust to naively chosen set of sampling probabilities, or carefully designed probabilities?
- On dense reward benchmarks, it's not clear whether the performance gain comes from over-sampling of high-rewarding states or SSET. What would be the performance of sampling minibatches with priorities correlated reward scales?
- Plots should be consistent by reporting both mean and standard deviations. Currently some figures report standard deviations but some are not. Moreover, why does Figure 12 report the std over each run but not across multiple runs? It could help understanding the significance of results.
- Performance of SSET is not significant over baselines in MiniGrid domain as confidence intervals overlap with each other, making the claims not convincing enough. Conducting more supporting experiments on more complex domains (e.g., Car Racing environments) could help.

**Strengths And Weaknesses:**

Strengths
- Intuitive method for utilizing event states when they are available
- Extensive experiments on a range of tasks and domains

Weaknesses
- Inconsistency in plots for reporting results statistics
- Lack of analyss on the effect of additionally introduced hyperparameters (e.g., Event history length ($\tau$), Event sampling probabilities ($\eta_{i}$))
- Lack of statistical significance over baselines on MiniGrid domain

Overall
- This paper proposes an intuitive method for utilizing event states, but its focus on MiniGrid experiments where results are not significant and the lack of analysis on additionally introduced hyperparameters make the claims in the paper not convincing enough.

---

> ### Author Response · Authors · 2023-03-18
> **Response to Reviewer j8j6 (R2)**
>
> Thank you very much for your comments on the paper.  In response to your comments we have made the following changes:
>
> Comment: Investigation into the importance of history length…
>
> Response: As mentioned in the global response, in addition to the original Obstacle Course history length investigation in Figure 9, we have now also included history length ablations in the Mujoco domains (where events are based on rewards and therefore usually intermediate)  in Figure 13 and described at the end of Section 6.  Here we again see the general benefits of longer histories (as predicted by the theory) but that the effect is somewhat modest since the algorithm is generally robust to non-extreme tau values.
>
> Comment: Investigation into the sensitivity to event sampling probabilities…
>
> Response: As mentioned in the global response, in addition to the original investigation of $\eta_0$ (an ablation of the total sum of non-default table sizes) in Figure 5, we have now added an additional investigation of individual $\eta_i$ probabilities in the Obstacle Course domain (a domain with a  large number of tables).  In Figure 10 we now show comparisons of the original hand-tuned weights vs, weights that are all equal, and two extreme cases where a single table (key or goal) dominates the bulk of the weights. Here, we see that the performance is similar with the extreme sets showing some minor fluctuations at the end, but still performing better compared to using uniform experience replay.
>
> Comment: On dense reward benchmarks, it's not clear whether the performance gain comes from over-sampling of high-rewarding states or SSET. What would be the performance of sampling minibatches with priorities correlated reward scales?
>
> Response: We believe the new ablation studies of history length (Figure 13 from the first bullet here) show the benefits of SSET here.  In particular, only over-sampling the high reward states at $\tau=1$ (the teal lines in the Figures) is consistently either the worst or one of the worst performers. Therefore, the history component of SSET (which samples slower-reward states that lead to higher reward ones) is having an effect, not just over-sampling of some of the high-reward states.
>
> Comment: Plots should be consistent by reporting both mean and standard deviations. Currently some figures report standard deviations but some are not. Moreover, why does Figure 12 report the std over each run but not across multiple runs? It could help understanding the significance of results. And performance of SSET is not significant over baselines in MiniGrid domain as confidence intervals overlap with each other…
>
> Response: As mentioned in the global response, we apologize for the confusion here.  In the new manuscript, we have replaced the pure variance shading with a standard error confidence interval in all the learning curve plots.  As you can see now, the separation is much clearer in the charts and the results show statistical significance in all the crucial comparisons to baselines.  As for Figure 12 (now Figure 14), the Gran Turismo experiments require a significant amount of (scarce) computing resources, including PlayStations, so we only performed 5 runs per experiment in that section.  With only 5 runs, standard deviation across the runs is not particularly meaningful and we think showing all 5 individual curves is actually the most transparent way to present the results and the variance between runs can be seen by visually looking at the 5 similarly-colored performance measures.
>
> Thank you again for your suggestions.

---

### Review · Reviewer_Gpmo · 2023-03-05

**Summary Of Contributions:**

This paper proposes the use of event tables, special replay buffers filled with experience related to particular events happening in the environment, to speed up training in reinforcement learning algorithms. The paper presents SSET, an algorithm based on the idea of executing stratified sampling from the event tables, as well as a theoretical and empirical analysis of this algorithm in comparison to other off-policy approaches.

**Audience:**

Yes

**Broader Impact Concerns:**

I have no particular concerns about the broader impact of this work.

**Claims And Evidence:**

Yes

**Requested Changes:**

Results, especially in continuous control, would be more convincing if using an evaluation that allows to draw statistically meaningful conclusions about the performance of the presented algorithm compared to other baselines.
I recommend the authors to look into the evaluation procedures proposed by "Deep Reinforcement Learning at the Edge of the Statistical Precipice" (Agarwal et al., 2021), which are directly applicable to similar domains such as the ones from DeepMind Control Suite (it just requires normalized returns for meaningful aggregation).

**Strengths And Weaknesses:**

Strengths:
- The approach presented in the paper paper is sound and, as far as I know, novel and well-positioned with respect to existing work.
- The method seems to be easily implementable and potentially very helpful for applying reinforcement learning to any domain in which identifying successful events is easy but designing a reward function is harder.
- The empirical analysis of the presented method is quite thorough and enjoyable. As an example, I particularly liked the analysis in Section 5.6, showing in an easily understandable example the impact of the decisions around event selection.

Weaknesses:
- I am concerned about the lack of error bands on some of the plots concerning the empirical evaluation of the method (e.g., Figure 7-8-9), especially given the noisy nature of the episode return curves.
- More generally, overlapping error bands in other plots (e.g., Figure 11) undermine the ability to understand whether SSET performs better than other prioritization approaches in a statistically significant way.
- One drawback of the proposed method is its reliance on manually-selected events. Despite there is already some discussion on the robustness of the method to bad event selection, I think the paper would benefit from extended discussion on what would the difficulties be in designing a system that automatically learns how to identify such events.

---

> ### Author Response · Authors · 2023-03-18
> **Response to Reviewer Gpmo (R3)**
>
> Thank you very much for your comments on the paper.  In response to your (and R2’s) concerns about the lack of error-bands and overlapping error bands, we have replaced the original pure-variance bands with a statistically valid confidence term, standard error.  We have included std. error bands on all the learning curve plots and, as you can now see, there is now a significant (non-overlapping) gap between baselines and SSET variants.
>
> As for highlighted the difficulties of directly learning events, yes we have included an updated paragraph in Section 8 that reads as follows:
>
> Beyond these guidelines for using user-specified events, the topic of learning a helpful set of events online is an open, but intriguing question.  The difficulty in identifying events with no background knowledge is that by Definition 4 in the Appendix, events correspond to states that are most likely on optimal trajectories, so in the extreme case knowing the perfect set of events means knowing the optimal policy's state distribution.  However, in cases where some model information is known, subgoal discovery methods could be used to propose events.  One could also over-specify a large number of events and attempt to learn the weights ($\eta$ and $\kappa$) online, though the current work does not guarantee the convergence of such a meta-learner.
>
> Thank you again for your suggestions.

---

### Decision · Action_Editors · 2023-04-10

**Recommendation:** Accept as is

**Comment:**

The paper is easy to read and the idea useful. The paper can be accepted mostly as is.

However, I have one very minor request (two sentences at most). I would ask for the camera ready version that you give a couple of sentences of explanation for the choice of hyperparameters in Section A.2. In A.3, you justify that most hyperparameters were set to the previous implementation. But for Section A.2, there is no such explanation. Tuning hyperparameters for your approach can cause misleading results. I am giving the benefit of the doubt that you have not done so here, but you should explain to the reader where these came from (e.g., why epsilon = 0.3?).

**Audience:**

Yes, a broad swath of the RL community would be interested in improved mechanism for replay.

**Claims And Evidence:**

The paper is clearly written with both empirical and theoretical supporting evidence. The paper generally does not overclaim, and there was a reasonable attempt to characterize statistical significance with t-tests.

---

> ### Author Response · Authors · 2023-05-05
> **Camera ready version uploaded**
>
> Dear Editors and Reviewers,
>
> Thank you all for your valuable comments, suggestions and for accepting our submission. We are delighted!
>
> In case there wasn't an automatic notification, we uploaded the camera ready version of the paper last Sunday after adding a couple of lines on the hyperparameters selection (as suggested by the action editor). Please let us know if there is anything else to be done from our end.
>
> Thanks,
> Authors